# Recursive Binding on a Budget: Subspace Carving in Order-$p$ Tensor Memories

**Travis Pence** [1]   **Daisuke Yamada** [1]   **Vikas Singh** [2][1]

## Abstract

Tensor Product Representations provide the structural fidelity required for symbolic reasoning in models but suffer from *exponential* dimensionality growth when encoding deep recursive structures. Conversely, Vector Symbolic Architectures maintain *constant* dimensionality but sacrifice capacity and fidelity due to noisy compression via superposition. In this work, we propose **Orthogonal Subspace Carving (OSC)**, a memory architecture that binds *fillers* to *roles* by projecting onto the null space of the role basis before aggregating into a fixed order-$p$ tensor. OSC uses projections to enforce geometric orthogonality between bound structures within a *static* memory trace. We show that this mechanism decouples the tensor order from the structural depth, enabling deep recursive binding within a *constant* memory footprint. By performing retrieval via recognition, this construction allows for component vectors that are *orders of magnitude* smaller than the memory tensor, giving superior memory efficiency in settings involving high superposition. We also show that TPR is a special case of binding in Clifford algebra, and give a Clifford formulation of OSC.

## 1. Introduction

Human reasoning is distinguished by its ability to perform structured operations over symbolic representations (Newell, 1980; 1982; Marcus, 2001). Standard neural network models struggle to systematically generalize to novel compositions of known structures (Dziri et al., 2023; Li et al., 2023; Kim & Linzen, 2020; Keysers et al., 2020). Bridging this "neuro-symbolic gap" requires representations that can support rich, recursive structure but are compatible with the gradient-based optimization of continuous vector spaces (Kanerva, 2009; Kleyko et al., 2022).

[1]Department of Computer Science, University of Wisconsin-Madison, United States [2]Department of Biostatistics and Med. Info., University of Wisconsin-Madison, United States. Correspondence to: Travis Pence <tnpence@wisc.edu>.

*Proceedings of the 43$^{rd}$ International Conference on Machine Learning*, Seoul, South Korea. PMLR 306, 2026. Copyright 2026 by the author(s).

An important line of work addressing the challenge above is the Tensor Product Representation (TPR) (Smolensky, 1990). As we will discuss in more detail shortly, TPRs provide a mathematically rigorous method for binding variables ("fillers") to structural "roles" via the outer product operation. This explicitly constructs a representation that preserves perfect orthogonality between bindings, and allows exact retrieval of constituents without interference (i.e., perfectly). However, the dimensionality of a TPR scales *exponentially* with the depth of the structure. For example, a tree of depth $k$ needs a tensor space of order $d^k$, making exact deep recursive binding intractable (Soulos et al., 2023). While recent sparse approximations can mitigate this scaling, they suffer from significant performance degradation on deep structures (Soulos et al., 2024).

Vector Symbolic Architectures (VSAs) offer a compressed alternative (Kanerva, 2009). VSAs employ "reduced" binding operations, such as circular convolution or element-wise multiplication, that map the tensor product back into the fixed-dimensional space of the component vectors (Kleyko et al., 2022). This fixed dimensionality makes VSAs efficient. It allows encoding complex structures within a *constant* memory footprint. But, there is a trade-off. In VSAs that use approximate inverses, non-orthogonal noise *accumulates* with depth. Further, as memory becomes crowded (more superposition), SNR degrades, eventually impacting reliable retrieval (Gosmann & Eliasmith, 2019; Plate, 1995).

Our **contributions** are: **(a)** OSC, a novel memory architecture that decouples component vector dimension from memory capacity via a projection-based binding mechanism; **(b)** a large reduction in memory footprint for high-superposition tasks. We get robust retrieval with component vectors orders of magnitude smaller than the memory space; **(c)** near-lossless retrieval accuracy in high-superposition regimes, enabled by sub-linear filler scaling that allows for larger superposition memory spaces within a smaller total memory footprint without TPR's exponential dimensionality cost. We do extensive empirical verification against VSA and in standard downstream applications. The code is available at `https://github.com/vsingh-group/OrthogonalSubspaceCarving`.

**Conflict of Interest Disclosure.** There are no conflicts of interest.

## 2. Preliminaries

We first give a short overview of Tensor Product Representations (TPRs) and Vector Symbolic Architectures (VSAs).

**Spaces.** VSAs/TPRs operate with three specific types of "spaces", namely, **filler**, **role**, and **memory**. A *filler* represents the atomic content, concept, or object being stored (e.g., "blue", "dog", "5"). A *role* defines the structural slot or attribute associated with that content (e.g., "color", "animal", "magnitude"). The *association* of a role(s) to a filler creates an instance of that concept within the memory, allowing hierarchical data to be represented. See Figure 1.

**Definition 2.1** (VSA/TPR Filler). The **filler space** is the hyperspace $\mathcal{V}$, usually $\mathbb{R}^d$, $\mathbb{C}^d$, or $\{0,1\}^d$. A specific **filler** $f \in \mathcal{V}$ is a single vector in the filler space.

**Definition 2.2** (VSA/TPR Context). Similarly, the **role space** is also $\mathcal{V}$. A specific **role** $r \in \mathcal{V}$ is a single vector. A **context** $\mathcal{C}$ is just a collection of roles $r_1, r_2, \ldots, r_c \in \mathcal{V}$.

VSAs and TPRs use three main manipulation operations.

1. **Binding ($\otimes$):** This operator *associates* the role with the filler, creating the bound object $T = f \otimes r$.
2. **Unbinding ($\oslash$):** This operator uses the role to *retrieve* the filler from the object, $(f \otimes r) \oslash r = f$.
3. **Bundling ($\oplus$):** This operator superposes multiple bound objects into one memory space, $M = T_1 \oplus T_2 \oplus \cdots \oplus T_k$. This is almost always simple addition.

### 2.1. Tensor Product Representations (TPRs)

In TPRs, the memory space is *not* the same as the role and filler space but instead is a higher-order tensor.

**Definition 2.3** (TPR Binding). The binding operator is simply the tensor product. Given a filler $f \in \mathbb{R}^d$ and a role $r \in \mathbb{R}^d$, the bound object $T$ is a matrix in $\mathbb{R}^{d \times d}$:

$$T = f \otimes r = fr^\top \tag{1}$$

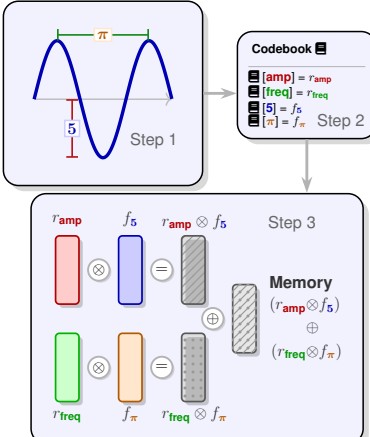

*Figure 1.* Raw signal attributes are identified (Step 1) and mapped to vectors using a codebook (Step 2). These are then bound to their corresponding roles and superposed to form the memory (Step 3).

So, when we associate 1 role to a filler, the memory space is an order-2 tensor. When binding $|\mathcal{C}|$ roles to a filler, the memory space is a order-$(|\mathcal{C}|+1)$ tensor with $d^{|\mathcal{C}|+1}$ entries. This grows exponentially with context size.

**Definition 2.4** (TPR Unbinding). Unbinding is the inner product *contraction* of the bound object with the role vector:

$$\hat{f} = Tr \tag{2}$$

**Orthogonality and Superposition.** A key property of TPRs is that if the role vectors are mutually orthogonal, superposition is *lossless*. That is, multiple filler-role bindings can be bundled into a single memory matrix $M$ without *any interference* between terms. Let roles $r_1, r_2$ be orthogonal.

**Example 2.5.** *For memory $M = f_1 r_1^\top + f_2 r_2^\top$, retrieval is exact:*

$$Mr_1 = f_1(r_1^\top r_1) + f_2(r_2^\top r_1) = f_1(1) + f_2(0) = f_1 \tag{3}$$

### 2.2. Vector Symbolic Architectures (VSAs)

TPRs guarantee perfect retrieval via orthogonality but memory size scales exponentially (with context size). VSAs instead use compressed, dimensionality-preserving operations. Thus, VSAs can be viewed as an approximation of TPRs. Basically, the tensor product is "collapsed" back into the original vector space $\mathcal{V}$, trading exactness for a fixed memory size.

**Definition 2.6** (VSA Binding). The binding operator is a map $\otimes : \mathcal{V} \times \mathcal{V} \to \mathcal{V}$ that *preserves* dimensionality. For a filler $f \in \mathbb{R}^d$ and a role $r \in \mathbb{R}^d$, the bound object $T$ remains a vector in $\mathbb{R}^d$:

$$T = f \otimes r \in \mathbb{R}^d \tag{4}$$

**Definition 2.7** (VSA Unbinding). Unbinding $\oslash : \mathcal{V} \times \mathcal{V} \to \mathcal{V}$ also preserves dimensionality. Given a bound object $T = f \otimes r$, unbinding with the role gives a version of the filler:

$$\hat{f} = T \oslash r \tag{5}$$

In many architectures, this operation is exact for a *single* pair ($\hat{f} = f$). But when unbinding with a superposition of multiple pairs, unbinding gives $f$ plus a pseudo-random noise term (from interference of the other stored pairs).

$$(f_1 \otimes r_1 + f_2 \otimes r_2) \oslash r_1 = f_1 + \underbrace{(f_2 \otimes r_2) \oslash r_1}_{\text{noise}} \tag{6}$$

### 2.3. Recognition, Recall, and Clean-up Memories

Symbolic memory systems can use two distinct query modes. **Recall** reconstructs content from a cue: "What is stored at this location?" In contrast, **recognition** tests a hypothesis: "Is this specific binding present?" Recognition compares a candidate binding against the memory and

returns a similarity score. Many systems combine unbinding with a **clean-up memory**. This is a codebook of valid symbols. After unbinding yields a noisy vector, the system can identify the nearest codebook entry. This effectively converts recall into recognition: testing each vocabulary item and selecting the best match.

## 2.4. Projection Operators

For a subspace $S \subset \mathbb{R}^d$ with orthonormal basis matrix $B$, the orthogonal projection onto $S$ is $P_S = B^T B$. The complementary projection onto the orthogonal complement is $P_S^\perp = I - B^T B$. $P_S^\perp$ is idempotent ($P_S^\perp P_S^\perp = P_S^\perp$), and $P_S P_S^\perp = 0$ (projections are disjoint).

# 3. Structural Challenge of Symbolic Memory

Consider the task of storing 1000 parse trees, each of depth 5, in a symbolic memory. A standard TPR needs a memory tensor of order $d^5$ for even modest dimension $d = 128$. This needs $3 \times 10^{10}$ parameters. VSAs will keep memory at $d = 128$, which is good. But superposing 1000 bindings drives the SNR to $\sqrt{128/1000} \approx .358$ (close to noise floor).

**The challenge.** To support symbolic reasoning, a memory system must accommodate deep recursive structures, allow for superposition of multiple structures within a single trace, and maintain high discrimination capabilities at query time. These requirements work against one another.

## 3.1. The Memory Triangle

The tension we describe above manifests as a three-way trade-off between structural fidelity, memory footprint, and superposition capacity that we can check quickly.

**Perfect retrieval $\rightarrow$ exponential growth.** TPRs place each binding in a distinct tensor dimension. This gives zero interference. A tree of depth $k$ requires $O(d^k)$ space which is intractable for even modest depth.

**Fixed footprint $\rightarrow$ accumulated interference.** VSAs compress all bindings into $O(d)$ dimensions. Superposing $N$ items introduces interference scaling as $O(\sqrt{N/d})$. The capacity degrades as memory fills.

All approaches must occupy some region of the triangle. Trade-offs are unavoidable. The question is about *which* trade-offs yield favorable scaling given the constraints.

This trade-off can also be cast geometrically. If $N$ unit-norm memory traces are embedded in a $D$-dimensional space, the average squared coherence $\mu^2 = \frac{1}{N(N-1)} \sum_{i \neq j} |\langle T_i, T_j \rangle|^2$ is bounded below by the Welch bound (Welch, 1974)

$$\mu^2 \geq \frac{N - D}{D(N - 1)} \quad \text{whenever } N > D. \tag{7}$$

Once the number of stored items exceeds the effective dimension, cross-talk cannot be driven to zero. The relevant design question is therefore not whether interference can be eliminated, but how an architecture allocates effective dimension among context complexity, vocabulary size, and superposition capacity. OSC takes advantage of an effective dimension of $d^p$ for the memory tensor while keeping per-filler storage at $p \cdot d$.

## 3.2. Recognition and Local structure can suffice

To navigate this trade-off, we must re-evaluate what is strictly necessary for symbolic processing.

**Recognition versus reconstruction.** Most symbolic reasoning systems operate over fixed vocabularies: parse trees use a grammar, programs use a type system. This prior knowledge shifts the retrieval task in an important way. Rather than reconstructing an arbitrary vector from a noisy trace (the hard problem), the system needs only to determine *which symbol from the codebook is most strongly present*. This recasts the goal from generative reconstruction to discriminative recognition. Even VSAs and TPRs with algebraic unbinding rely on this principle. After unbinding, a cleanup memory maps the noisy result to the nearest vocabulary item. The "exact" unbinding is discarded in favor of vocabulary search. If recognition is unavoidable, why pay the cost of maintaining exact unbinding?

**Only Local precision.** Structural guarantees need not be universal. If a system can ensure that fillers relevant to a *specific query context* are geometrically distinct, it can discriminate well – even if fillers in different contexts are only statistically independent. This suggests a design principle: enforce orthogonality *locally* within a context, accept quasi-orthogonality *globally* across contexts. When querying "subject of this sentence", it is fine if candidate subjects are distinguishable from each other. They need not be orthogonal to a verb of an unrelated sentence stored earlier.

**A Geometric Reformulation?** These perspectives suggest a shift in how we approach the binding problem. Rather than expanding the dimension to accommodate all possible structures, we can treat structure as a geometric constraint **within a fixed space** via an operation that carves out projected regions for specific roles. Next, we introduce a special memory architecture that realizes these principles.

# 4. Orthogonal Subspace Carving (OSC)

Rather than assigning bindings to explicit coordinate slots, our approach will work by defining where information is *not* allowed to go. Each context is associated with a geometric subspace that acts as a "forbidden" region. Bindings are then placed in the orthogonal complement, this is the space that remains after carving out the context.

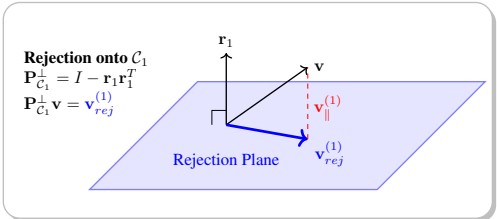 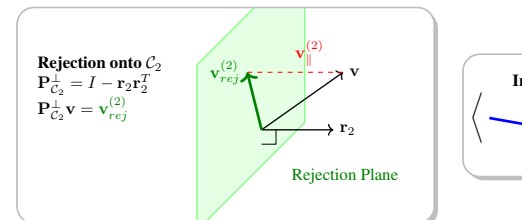

*Figure 2.* Visualization of the rejection process. The vector $\mathbf{v}$ is projected onto orthogonal complements of $\mathcal{C}_1$ and $\mathcal{C}_2$, resulting in nearly orthogonal rejection vectors. The more vectors in a filler, the less likelihood that all rejection vectors will align.

**High level Principle.** The reversal described above is key. Standard approaches mark storage locations explicitly: "role 1 goes in dimension 7". OSC instead says: "this context occupies dimensions $\{7, 42\}$. So, all bindings must avoid them." The binding lives in the remaining $d - 2$ dimensions, orthogonal to the context by design. This exclusion ensures that bindings under different contexts occupy *disjoint subspaces* without requiring coordination, and separation is automatic. Structure emerges from what is carved away.

### 4.1. Representational Ingredients.

The architecture operates within a fixed-dimensional ambient space $\mathbb{R}^d$. Unlike conventional symbolic architectures (fillers as single vectors), we construct fillers as *structured objects* with multiple components.

**Definition 4.1** (OSC Filler). Let $p$ be a positive integer denoting the order. The filler space $\mathcal{F}$ is defined as the Cartesian product of $p$ vector spaces (each of dimension $d$):

$$\mathcal{F} = (\mathbb{R}^d)^p \tag{8}$$

A specific filler $\mathbf{f} \in \mathcal{F}$ is an ordered tuple of $p$ vectors:

$$\mathbf{f} = (v_1, v_2, \ldots, v_p) \tag{9}$$

where each component vector $v_i \in \mathbb{R}^d$ for $i \in \{1, \ldots, p\}$.

Contexts depart from conventional one-to-one role-vector mappings. Instead of associating each role with a single vector, we associate each context with a subspace that can accommodate multiple roles simultaneously.

**Definition 4.2** (Context). A context $\mathcal{C}$ is associated with a subspace $S_{\mathcal{C}} \subset \mathbb{R}^d$ spanned by a set of $k$ mutually orthogonal directions $\{u_1, u_2, \ldots, u_k\}$.

Orthogonality is required only among directions within a single context. Different contexts need not share any geometric relationship. This realizes the local versus global principle from §3: structural precision within each context but independence across contexts.

### 4.2. The Carving Operator

To enforce the exclusion principle, we define a projection that removes any overlap with the context subspace.

**Definition 4.3** (Subspace Carving Operation). Let $\mathcal{C}$ be a context with basis vectors $\{u_1, \ldots, u_k\}$. We stack these as rows to form the basis matrix $B_{\mathcal{C}} \in \mathbb{R}^{k \times d}$. The carving operator $P_{\mathcal{C}}^{\perp} \in \mathbb{R}^{d \times d}$ projects onto the null space of $B_{\mathcal{C}}$:

$$P_{\mathcal{C}}^{\perp} = I_d - B_{\mathcal{C}}^T B_{\mathcal{C}} \tag{10}$$

This operation removes any component of a vector that overlaps with the context's directions. For any vector $v \in \mathbb{R}^d$, the result $P_{\mathcal{C}}^{\perp} v$ is guaranteed to be orthogonal to all $u_i$, ensuring the processed vector lies entirely outside the context subspace $S_{\mathcal{C}}$ as seen in Figure 2. We provide motivation for the rejection in Appendix C.

### 4.3. Binding Operation

To bind a filler to a context, we apply the carving operator to each component independently. Then, we fuse the processed components into a higher-order structure.

**Definition 4.4** (Context Binding). For a filler $\mathbf{f} = (v_1, \ldots, v_p)$ and context $\mathcal{C}$, the bound object $T_{\text{bound}}$ is the order-$p$ tensor product of projected/normalized components:

$$T_{\text{bound}} = \bigotimes_{i=1}^{p} \left( \frac{P_{\mathcal{C}}^{\perp} v_i}{\|P_{\mathcal{C}}^{\perp} v_i\|} \right) \tag{11}$$

By normalizing, the tensor magnitude reflects alignment rather than scale. Each $v_i$ is first projected away from context, then normalized. Then, it is combined via the tensor product. We get an order-$p$ tensor $T_{\text{bound}} \in (\mathbb{R}^d)^{\otimes p}$ that encodes the filler while residing fully in $S_{\mathcal{C}}$'s complement.

### 4.4. Superposition and Memory Formation

The memory $\mathbf{M}$ is constructed by the superposition of such bound tensors using standard addition (like TPRs, VSAs):

$$\mathbf{M} = \sum_j \mathbf{T}_{\text{bound}}^{(j)} \tag{12}$$

The enforcement of separation takes place entirely during the binding stage, not at storage. Because each $T_{\text{bound}}^{(j)}$ is orthogonal to its respective context subspace and contexts define disjoint subspaces, the bindings naturally separate in the memory space. This makes memory formation order-independent (similar to existing architectures). Arbitrary number of bindings coexist using simple superposition.

## 4.5. Querying is Recognition

In OSC, we support recognition, not recall. To query the memory, we construct a candidate binding and measure its similarity against the stored memory.

**Definition 4.5** (Recognition Score). To test whether a candidate filler $\mathbf{g} = (v_1, \ldots, v_p)$ is associated with context $\mathcal{C}$, construct the candidate tensor $T_{\text{cand}}$ using the same binding mechanism:

$$T_{\text{cand}} = \bigotimes_{i=1}^{p} \left( \frac{P_{\mathcal{C}}^{\perp} v_i}{\|P_{\mathcal{C}}^{\perp} v_i\|} \right) \tag{13}$$

The recognition score is the Frobenius inner product:

$$\text{Score}(\mathbf{g}, \mathcal{C}) = \langle \mathbf{M}, T_{\text{cand}} \rangle_F \tag{14}$$

When the candidate matches a stored binding, the score is close to 1. When it does not match, the score is close to zero. The score yields a unit response for stored items and a zero-mean response for interference, with a variance that is suppressed as the tensor-order or dimension increases.

**Theorem 4.6** (Interference Scaling). *Let* $\mathbf{M}$ *be a memory superposition of* $N$ *bindings constructed from independent, isotropic random fillers. For a query matching a stored target, the retrieval score* $S$ *satisfies* $\mathbb{E}[S] = 1$. *The interference noise* $I$ *arising from the* $N - 1$ *other bindings has zero mean,* $\mathbb{E}[I] = 0$, *and a standard deviation that scales as:*

$$Std(S) = \sqrt{\underbrace{Var(Signal)}_{0} + Var(I)} = \sqrt{\frac{N-1}{d^p}} \tag{15}$$

*Proof.* See Appendix B for the full derivation. $\square$

**Retrieval via vocabulary search.** To retrieve the filler associated with a context, we evaluate all candidates from the vocabulary. For each filler in the codebook, we can construct its binding with the query context and calculate the score. Then, we select the candidate with the highest score. This trades algebraic unbinding for geometric interference control, and is favorable in high-superposition regimes.

## 4.6. Loss of Exact Unbinding

A structural consequence of this design is that exact algebraic unbinding is impossible. Because $\mathbf{P}_{\mathcal{C}}^{\perp}$ is a projection matrix, it is singular. Information in the subspace spanned by $\mathcal{C}$ is discarded. So, the memory allows queries of the form *"Is X stored here?"* but not *"What is stored here?"* without iterating through the vocabulary. However, as discussed, even architectures with algebraic unbinding (VSAs, TPRs) rely on cleanup memories that perform vocabulary search. Since vocabulary search is unavoidable, the inability to unbind exactly is not a functional disadvantage. In fact,

for VSAs where the unbinding operation is the adjoint of binding (HRR, MAP, HLB) retrieval via recognition yields mathematically identical rankings to standard unbinding-based retrieval, since the adjoint property guarantees the argmax over the codebook is equal (Appendix D.2).

We call the formulation **Orthogonal Subspace Carving** (OSC). It enforces structural distinctions through geometric exclusion: contexts carve forbidden subspaces. This shifts the goal from reconstruction to discriminative recognition.

## 4.7. Connection to Clifford algebra

TPR and OSC connect naturally to Clifford algebra, which has attracted growing interest in machine learning, e.g., rotor-based linear layers (Pence et al., 2025), general Clifford network layers (Ruhe et al., 2023), and characterizations of network weights (Pilanci, 2024). We show TPR admits a strict generalization within Clifford algebra. In the Clifford construction, instead of requiring roles and fillers to be supported on disjoint basis vectors, which recovers TPR, they need only be orthogonal to each other for exact recovery of fillers, reducing the tensor memory from $d^{|C|+1}$ to $\binom{d}{|C|+1}$.

OSC admits a natural description in this same framework. The context is encoded as a single algebraic object representing the subspace itself, rather than as a matrix describing it. The carving operator $P_{\mathcal{C}}^{\perp}$ then becomes a geometric rejection that strips away whatever part of a vector lies inside this subspace, leaving only the component orthogonal to it. Fillers are likewise lifted from tuples of vectors to subspaces in their own right, and binding combines a filler subspace with the orthogonal complement of the context. The same orthogonality between fillers and roles that makes the TPR generalization exact is the geometric relationship that OSC enforces through subspace carving. See Appendix C for the TPR Clifford algebra generalization and the OSC Clifford equivalent.

# 5. Scalability

We introduced OSC's binding mechanism, which relies on contexts defining forbidden subspaces. In this section, we talk about the scalability of roles and fillers.

## 5.1. Procedural Context Generation.

Contexts no longer need a global lookup table and can instead depend solely on the abstract roles used to define them, i.e., they can be generated procedurally on demand. The mechanism is simple: combine the symbolic labels of all constituent roles (e.g., "subject", "sentence_5", "parse_tree") into a unique string, hash this string to produce a random seed, and use the seed to generate the context basis (mu-

tually orthogonal basis vectors $\mathbf{B}_\mathcal{C}$). More precisely, for a context $\mathcal{C}$ defined by role labels $\{\ell_1, \ldots, \ell_k\}$:

**Step 1** Concatenate labels: $s = \text{concat}(\ell_1, \ldots, \ell_k)$
**Step 2** Hash to seed: $\text{seed} = \text{hash}(s)$ (e.g., SHA-256)
**Step 3** Generate random matrix: $G \in \mathbb{R}^{k \times d}$ using seed
**Step 4** Orthonormalize: $B_\mathcal{C} = \text{QR}(G)$

**Context independence of Memory Footprint.** Because this process is deterministic, the exact orthogonal subspace can be recreated whenever the context is defined. There is no need to store the basis vectors themselves, decoupling the number of representable contexts from the memory constraints. Increasing complexity (associating more roles with a filler) costs nothing in terms of memory. This is in sharp contrast to TPRs, where the dimensionality of the representation scales exponentially.

## 5.2. Filler Representation Efficiency

The multi-component filler structure provides a second advantage: vocabulary cost grows sub-linearly with memory capacity.

**VSA.** In VSAs, fillers and memory occupy the same $d$-dimensional space. A filler is $d$ parameters, so the cost is:

$$N_{\text{VSA}} = d + |\mathcal{V}| \cdot d = d(1 + |\mathcal{V}|) \quad (16)$$

Each filler costs as many parameters as the entire memory.

**OSC Advantage.** OSC decouples filler dimension from memory dimension. The memory tensor has effective dimension $d_1 = d^p$, but fillers remain in $(\mathbb{R}^d)^p$, requiring only

$p \cdot d = p \cdot d_1^{1/p}$ parameters each:

$$N_{\text{OSC}} = d_1 + |\mathcal{V}| \cdot p \cdot d_1^{1/p} \quad (17)$$

Vocabulary cost grows as $\mathcal{O}(d_1^{1/p})$ which is sub-linear in memory capacity. We show the compounding effect this has on total memory in §6.

## 6. Experiments

**Goals.** We evaluate the scalability and efficiency of OSC through the standard synthetic benchmarks and model integration tasks. The main goals are: **(G-1)** *Juxtapose memory capacity with total storage efficiency*, demonstrating that OSC performs better in terms of total relative size, maintaining a drastically smaller footprint as the number of stored bundles increases. **(G-2)** Validate that the *construction is learnable* by integrating OSC into models for common VSA applications.

**Scope of Baselines.** We exclude TPRs from most of our experiments for structural reasons. Indeed, the entire class of VSAs exists because TPRs become computationally intractable for high context sizes. In contrast, the performance of OSC is invariant to context complexity but sensitive to the degree of superpositon. We compare OSC to TPRs where TPRs remain feasible at a role depth of 1.

We evaluate OSC against 14 different VSAs, most examined in two recent survey papers (Schlegel et al., 2021; Kleyko et al., 2022). The classical baselines are Holographic Reduced Representations (HRR) (Plate, 1995), Fourier-HRR (FHRR) (Plate, 2003), Multiply-Add-Permute variants (MAP-I, MAP-C, MAP-B) (Gayler, 1998), Binary Spat-

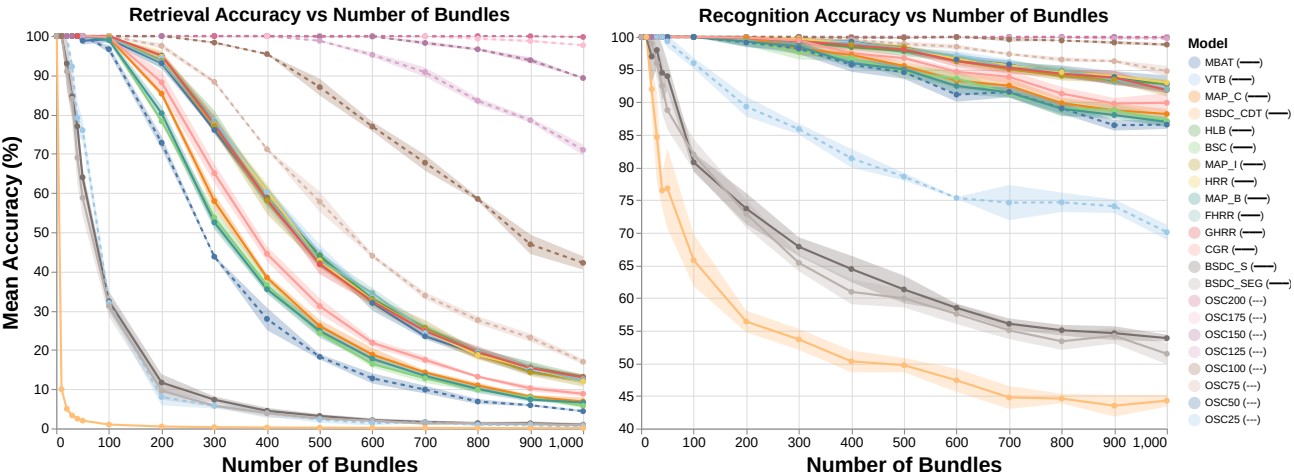

*Figure 3.* Performance comparison of 14 VSAs at $d = 4096$ and OSC across $d = 25, 50, 75, 100, 125, 150, 175,$ and $200$ for $p = 2$ at increasing bundle counts with unique fillers and role depth 1. OSC outperforms all VSAs on both retrieval and recognition tasks. Note the difference in $Y$ axes between graphs. Standard deviation is denoted by the shaded regions. We provide comparisons for other role depths and VSA dimensions in Appendix D.

ter Codes (BSC) (Kanerva, 1996), and Binary Sparse Distributed Representation variants (BSDC-S, BSDC-CDT) (Rachkovskij, 2002). The more modern baselines are: BSDC-SEG (Laiho et al., 2015), Matrix Binding of Additive Terms (MBAT) (Gallant & Okaywe, 2015), Vector-Derived Transformation Binding (VTB) (Gosmann & Eliasmith, 2019), Cyclic Group Representation (CGR) (Yu et al., 2022), Hadamard-derived Linear Binding (HLB) (Alam et al., 2024), and Generalized-HRR (GHRR) (Yeung et al., 2024).

| Model | Retrieval Acc. | Recognition Acc. | Total Memory |
|---|---|---|---|
| Best VSA ($d = 4096$) | $13.18 \pm .58$ | $93.00 \pm .51$ | 4100096 |
| Best VSA ($d = 8100$) | $35.88 \pm .40$ | $98.04 \pm .26$ | 8108100 |
| Best OSC ($d = 200$) | $\mathbf{99.77 \pm .09}$ | $\mathbf{100 \pm .00}$ | 440000 |

*Table 1.* Accompanying table to Figure 3. Retrieval and recognition accuracy for the best-performing VSA at 2 different dimensions and OSC at 1000 bundles. With less memory, OSC outperforms the best VSA on retrieval and recognition tasks.

**Why Synthetic Evaluation?** Our core scalability experiments rely on synthetic benchmarks, following the standard evaluation for novel VSA architectures (Schlegel et al., 2021; Gosmann & Eliasmith, 2019; Alam et al., 2024). This design choice is grounded in practice: in common VSA use cases, vectors are often fixed randomly generated codebooks (not learned). Thus, these simulations are not just approximate theoretic limits but reflect expected performance.

## 6.1. Memory Capacity and Scalability

**Experimental Setup.** We generate unique roles for every position in the memory. In the memory analysis, we exclude role storage costs for VSAs since this varies by application - from just 1 or 2 roles to thousands. While this underestimates the true VSA footprint, the clear difference between VSAs and OSC remains. We assess recognition and recall.

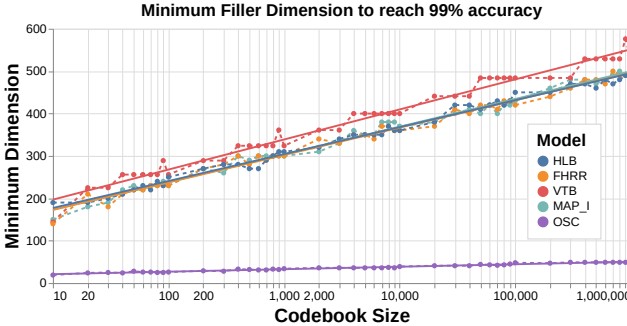

*Figure 4.* Minimum filler dimension required to achieve 99% accuracy. A dimension was selected only if mean accuracy across 10 trials exceeded 99% at that dimension but fell below 99% at the next lowest dimension. The dotted line tracks the actual dimension points, while the solid line represents the best fit. OSC exhibits the same favorable logarithmic scaling with codebook size as VSAs.

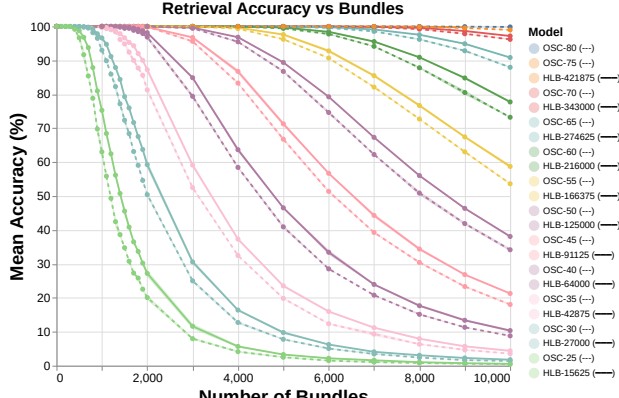

| Model | Retrieval Acc. | Recognition Acc. | Total Memory |
|---|---|---|---|
| Largest VSA ($d = 421875$) | OOM | OOM | 4219171875 |
| Best VSA ($d = 343000$) | $97.19 \pm .10$ | $\mathbf{100 \pm .00}$ | 3430343000 |
| Worst VSA ($d = 15625$) | $0.56 \pm .09$ | $80.95 \pm .22$ | 156265625 |
| OSC ($d = 80$) | $\mathbf{99.85 \pm .04}$ | $\mathbf{100 \pm .00}$ | 2912000 |

*Figure 5.* Performance of HLB at varying dimensions against OSC for $p = 3$. Identically colored runs share the same superposition memory size (not total memory). VSAs are slightly superior than OSC in memory capacity, but OSC's sub-linear scaling requires the VSA to have $1178\times$ OSC's total memory to achieve comparable retrieval accuracy. Table values correspond to 10000 bundles. Standard deviations (shaded regions) are present but negligible.

**Results.** Due to the sub-linear scaling of the filler vocabulary, OSC achieves orders-of-magnitude improvements in storage efficiency over VSAs, particularly in high-superposition regimes. While standard VSAs typically operate at $d = 4096$ or $d = 8192$ (with VTB necessitating $d = 8100$), Figure 3 and Table 1 demonstrates that OSC with $d = 200$ and $p = 2$ achieves 99.77% retrieval accuracy for 1000 bundles. This corresponds to a $9.3\times$ storage reduction compared to the best VSA at $d = 4096$ (13.18% accuracy) and an $18.4\times$ reduction compared to the baseline at $d = 8100$ (35.88% accuracy).

The efficiency gap widens significantly at higher levels of superposition. We selected HLB for this comparison, as it demonstrated the highest performance among VSAs in Figure 3. As shown in Figure 5, OSC at $d = 80$ and $p = 3$ maintains 99.85% accuracy for 10000 bundles. In contrast, maximizing HLB performance required scaling to $d = 343000$ to achieve 97.19% accuracy, resulting in a $1,178\times$ larger parameter footprint. While VSAs exhibit slightly higher capacity when holding memory size constant, this advantage is theoretical rather than practical. Even when HLB is reduced to $d = 15625$, yielding 0.56% accuracy, it remains $53.7\times$ larger than OSC. Due to the sub-linear scaling of the filler vocabulary, OSC's storage efficiency is orders of magnitude smaller in practice, directly supporting **G-1**.

**Influence of Codebook Size.** In previous experiments, the codebook size was fixed to match the number of bindings. However, retrieval performance is inherently dependent on the total size of the codebook, even if the additional fillers

| | TPR | | OSC ($p = 3$) | | Compression Ratio |
|---|---|---|---|---|---|
| $N$ | $d$ | Total Params | $d$ | Total Params | TPR/OSC |
| 50 | 44 | 4K | 13 | 4K | 1.00× |
| 100 | 61 | 10K | 16 | 9K | 1.10× |
| 500 | 137 | 87K | 27 | 60K | 1.45× |
| 1,000 | 197 | 236K | 34 | 141K | 1.67× |
| 5,000 | 451 | 2.5M | 59 | 1.1M | 2.25× |
| 10,000 | 649 | 6.9M | 76 | 2.7M | 2.55× |

*Table 2.* Minimum filler dimension required to achieve 99% accuracy. At $N = 10,000$, OSC has $76 \cdot 3 = 228$ parameters per filler, while TPR uses 649 parameters, leading to the OSC's 2.55× less parameter usage.

are not currently stored in memory. This occurs because a larger search space increases the probability that a random filler will, by chance, exhibit higher similarity to the noisy retrieved vector than the true target. This relationship was originally characterized by Plate (1995) and extended to other VSAs by Schlegel et al. (2021), who showed that codebook capacity scales exponentially with vector dimension. In Figure 4, we show that OSC shares this favorable scaling property: as the codebook size increases while superposition remains constant, the required filler dimension, that which contributes most to total memory, needs to grow only logarithmically to maintain accuracy. The number of bindings is fixed at $N = 10$. We provide further experiments in Appendix D.

**Comparison to TPRs** We compare OSC against TPRs where they remain feasible at a role depth of 1. Following the set-up in previous codebook size experiment, we find the minimum dimensions for TPR and OSC to reach 99% accuracy across varying levels of superposition. Table 2 shows that OSC achieves this with fewer parameters after a small amount of superposition, and the compression ratio increasingly favors OSC at scale as the filler dimension begins to dominate parameter count. Note TPR is only exact when the dimension is large enough for orthogonality.

**Inference Speed.** Each VSA has distinct binding and unbinding mechanisms, and the effort invested in low-level tuning would be uneven across baselines, which makes a comprehensive speed comparison difficult. We benchmark against HLB, the best-performing VSA from Figure 3, whose binding and unbinding are elementwise multiplication and division, the simplest amongst all VSAs benchmarked against.

We report retrieval timings with superposition memory held constant and amortized generation costs excluded. Because HLB's elementwise operations are simple, it is faster at small enough dimensions. However, as the representation scales through higher superposition, HLB's memory footprint dominates. At that point, even though OSC's perelement operations are more expensive, OSC becomes faster because the bottleneck is memory bandwidth, not compute. Table 3 reports wall-clock retrieval times and parameter

counts across three configurations. See Appendix D.4 for a FLOPs comparison.

### 6.2. Use for Extreme Multi-label Learning (XML)

We show that OSC, like other VSAs, is learnable in models. To validate this, we apply OSC to the task of Extreme Multi-label Classification (XML).

**Problem Definition.** XML is a classification setting where the output space consists of a massive number of classes ($L \geq 100000$), but the number of positive labels for any single input is sparse ($K \approx 10$). Standard neural network approaches (e.g., a final linear layer of size $d \times L$) scale poorly due to the size of the output space.

**Neuro-symbolic VSA Approach.** Ganesan et al. (2021) proposed a neuro-symbolic approach to solve XML by replacing the computationally expensive final layer with VSA operations. Instead of learning a classifier for every class, each class $k$ is assigned a fixed, random atomic vector $c_k$. The model is trained to regress a single superposition vector $S$ that represents the set of all active labels for the input.

| | | Speed (ms) | | | Parameter Count | | |
|---|---|---|---|---|---|---|---|
| | $N$ | 100 | 1,000 | 10,000 | 100 | 1,000 | 10,000 |
| **B1** | HLB | **0.068** | **0.071** | 0.180 | $414K$ | $4.1M$ | $41.0M$ |
| | OSC | 0.149 | 0.140 | **0.156** | $\mathbf{17}K$ | $\mathbf{132}K$ | $\mathbf{1.3}M$ |
| | HLB/OSC | 0.46× | 0.50× | 1.15× | 24.5× | 31.0× | 31.9× |
| **B2** | HLB | **0.071** | **0.196** | 1.189 | $4.0M$ | $40.0M$ | $400M$ |
| | OSC | 0.155 | 0.203 | **0.306** | $\mathbf{80}K$ | $\mathbf{440}K$ | $\mathbf{4.0}M$ |
| | HLB/OSC | 0.46× | 0.97× | 3.88× | 50.5× | 91.0× | 99.0× |
| **B3** | HLB | **0.261** | 1.345 | OOM | $42.6M$ | $422M$ | OOM |
| | OSC | 0.384 | **0.407** | **1.252** | $\mathbf{444}K$ | $\mathbf{647}K$ | $\mathbf{2.7}M$ |
| | HLB/OSC | 0.68× | 3.31× | NA | 95.9× | 653× | NA |

*Table 3.* Wall-clock retrieval time and parameter count for HLB vs. OSC across three benchmarks. **B1**: OSC ($d = 64, p = 2$) vs HLB ($d = 4,096$). **B2**: OSC ($d = 200, p = 2$) vs HLB ($d = 40,000$). **B3**: OSC ($d = 75, p = 3$) vs HLB ($d = 421,875$). $N$ denotes the number of superposed bindings. A ratio $> 1$ indicates OSC is faster or uses less memory.

**Efficient Training via Linearity.** A naive superposition of all $L$ classes would remain expensive. However, by defining two special role vectors—$p$ ("present") and $m$ ("missing"), we can exploit the linearity of VSAs to shift the compute complexity from the total number of classes $O(L)$ to the number of active classes $O(K)$.

The model is trained to give a target vector $S$ as:

$$S = p \otimes \underbrace{\left( \sum_{i \in Y} c_i \right)}_{\text{Active Labels}} + m \otimes \underbrace{\left( \sum_{j \notin Y} c_j \right)}_{\text{Inactive Labels}} \quad (18)$$

Calculating the second term (inactive labels) is normally expensive. However, because the sum of *all* class vectors

| DATASET | BIBTEX | | DELICIOUS | | MEDIAMILL | | EURLEX-4K | |
|---------|--------|--------|--------|--------|--------|--------|--------|--------|
| METRICS | nDCG | PSnDCG | nDCG | PSnDCG | nDCG | PSnDCG | nDCG | PSnDCG |
| VTB | **58.44±0.28** | **52.46±0.34** | 60.82±0.30 | 35.13±0.27 | **73.43±0.47** | **64.31±0.42** | 51.53±1.24 | 29.61±0.81 |
| MAP | 55.25±0.69 | 49.22±0.70 | 59.16±0.51 | 34.01±0.24 | 72.63±0.48 | 63.59±0.43 | 52.33±0.81 | 30.19±0.55 |
| HLB | 55.56±0.49 | 49.49±0.55 | 59.49±0.42 | 34.29±0.16 | 73.31±0.20 | 64.13±0.21 | 52.54±0.93 | 30.24±0.54 |
| OSC | 57.08±0.30 | 51.63±0.32 | **61.21±0.15** | **35.46±0.12** | 73.40±0.41 | 63.99±0.38 | **59.70±0.09** | **35.77±0.07** |

| DATASET | EURLEX-4.3K | | WIKI10-31K | | AMAZON-13K | | DELICIOUS-200K | |
|---------|--------|--------|--------|--------|--------|--------|--------|--------|
| METRICS | nDCG | PSnDCG | nDCG | PSnDCG | nDCG | PSnDCG | nDCG | PSnDCG |
| VTB | **75.91±0.39** | **53.99±0.41** | 63.02±0.15 | 10.14±0.07 | 77.28±0.15 | 55.21±0.13 | 38.27±0.10 | 7.67±0.03 |
| MAP | 68.42±1.45 | 45.10±0.99 | 61.67±0.38 | 10.26±0.11 | 75.08±0.14 | 53.48±0.11 | 34.01±0.15 | 6.81±0.03 |
| HLB | 71.31±0.34 | 47.37±0.32 | **63.88±0.50** | 10.45±0.14 | 78.01±0.14 | 55.84±0.12 | **38.80±0.46** | **7.78±0.09** |
| OSC | 71.59±0.13 | 46.15±0.13 | 62.73±0.14 | **10.82±0.05** | 78.91±0.11 | 56.62±0.11 | 37.28±0.08 | 7.34±0.04 |

*Table 4.* Retrieval performance (nDCG@5 (↑) and PSnDCG@5 (↑)) of VTB, MAP, HLB, and OSC across eight datasets. **Bold** indicates the best, while underlining denotes the second best. OSC ranked first in 43.75% of cases, surpassing VTB (37.5%) and HLB (18.75%). In terms of consistency, OSC and VTB both placed in the top two 68.75% of the time, followed by HLB at 62.5%. MAP did not rank in the top two. Architecture and hyperparameters are detailed in Appendix E.

$A = \sum_{k=1}^{L} c_k$ is constant and pre-computable, we can rewrite the target using the complement of the active set:

$$S = p \otimes \left( \sum_{i \in Y} c_i \right) + m \otimes \left( A - \sum_{i \in Y} c_i \right) \quad (19)$$

This formulation allows the target to be computed in $O(K)$ time. The network is then trained using a cosine-similarity loss that encourages the "present" component of the output to align with the active class vectors, and the "missing" component to differ.

**Results.** Similar to (Alam et al., 2024), we check OSC on 8 benchmark XML datasets (Bhatia et al., 2016) using normalized discounted cumulative gain (nDCG) and propensity-scored (PS) based nDCG (PSnDCG) as suggested by (Jain et al., 2016). Table 4 shows that OSC learns effectively here, yielding performance competitive with MAP, HLB, and VTB. This confirms that our binding mechanism preserves learnability in gradient-based systems, directly supporting **G-2**.

## 7. Related Works

VSAs and TPRs are extensively covered in §1 and §2.

**Matrix Memories and Outer Product Storage.** Distributed associative memories based on outer product storage emerged independently in the early 1970s (Kohonen, 2009; Anderson, 1972; Nakano, 2007). The pseudo-inverse extension, Optimal Linear Associative Memory (OLAM) (Kohonen & Ruohonen, 1973), replaces the correlation sum with the Moore-Penrose solution, achieving perfect retrieval up to the dimensionality limit. Hopfield networks (Hopfield, 1982) popularized this framework by introducing energy-based dynamics and stability analysis (McEliece et al., 2003; Baldi & Venkatesh, 1987). Modern work on dense associative memories (Krotov & Hopfield, 2016; Demircigil et al.,

2017) showed that higher-order interactions enable exponential storage capacity, connecting classical memory models to deep learning (Krotov & Hopfield, 2018). Recent analyses characterize capacity under various conditions (Lucibello & Mézard, 2024; Hu et al., 2024) and extend these models to kernelized settings (Wu et al., 2024).

**Conceptors and Gradient Projection Methods.** Conceptors (Jaeger, 2014; 2017) provide soft projection matrices for managing recurrent network dynamics, with Boolean operations enabling compositional control. Conceptor-Aided Backpropagation (He & Jaeger, 2018) applies this to continual learning by projecting weight updates onto null spaces of previous tasks. Orthogonal gradient projection methods pursue the same goal through different mechanisms: OWM (Zeng et al., 2019) maintains projectors from input representations, OGD (Farajtabar et al., 2020) stores gradient directions directly, and GPM (Saha et al., 2021) uses SVD on activations to find core gradient subspaces. Extensions include scaled gradient projection (Saha & Roy, 2023), class-level gradients (Chen et al., 2022), and adaptive orthogonal projection (Guo et al., 2022). These ideas have been applied to LLM fine-tuning via O-LoRA (Wang et al., 2023) and related methods (Tang et al., 2026).

## 8. Conclusion

We introduce Orthogonal Subspace Carving, a tensor memory architecture that projects fillers onto role null spaces before superposition, geometrically suppressing cross-talk between bound structures. OSC decouples context complexity from performance via procedural basis generation and offers significantly reduced memory footprint compared to traditional VSAs and TPRs due to the sub-linear scaling of the filler dimension. Finally, we demonstrate the compatibility of OSC with gradient-based models by validating its performance on XML datasets. See Appendix A for limitations and future work.

## Acknowledgments

We thank the anonymous reviewers for their constructive feedback. Authors were all partly supported by NIH R01AG092220.

## Impact Statement

This paper presents work whose goal is to advance the field of Machine Learning. There are many potential societal consequences of our work, none which we feel must be specifically highlighted here.

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

## A. Limitations and Future Work

As shown in Figure 8 and 9, the representation density of OSC is slightly lower than that of traditional VSAs. While OSC significantly outperforms VSAs in terms of total memory, VSAs are slightly better when superposition memory, not total storage, is the constraint.

As illustrated in Figure 1, in VSAs the bind operation produces a vector dissimilar to its constituents, while the bundle operation preserves similarity. OSC similarly preserves similarity during bundling, however unlike VSAs, it allows the binding operation to do so as well. In the extreme case, if a filler vector lies entirely within the orthogonal complement of the context, the rejection operation leaves the filler unchanged. Given that VSAs have established applications in information hiding (Alam et al., 2024), exploring the implications of this preserved similarity and mechanisms for encouraging dissimilarity in OSC remains a promising direction for future work.

## B. Retrieval Statistics

In this appendix, we derive the mean and variance of the retrieval score $S$ from Theorem 4.6. We assume that the component vectors $v \in \mathbb{R}^d$ are independent random vectors drawn from an isotropic Gaussian distribution, and that each binding uses an independently generated context.

### B.1. Score Definition

The retrieval score $S$ is defined as the Frobenius inner product between the memory tensor $M = \sum_{j=1}^{N} T^{(j)}$ and a candidate query tensor $T^{(q)}$.

Recall that each bound tensor $T$ is the order-$p$ tensor product of the projected, normalized filler components. Let $\tilde{v}_i^{(j)}$ denote the $i$-th component of the $j$-th filler after projection and normalization:

$$\tilde{v}_i^{(j)} = \frac{P_{C_j}^{\perp} v_i^{(j)}}{\|P_{C_j}^{\perp} v_i^{(j)}\|} \tag{20}$$

The pairwise inner product between two tensor terms $Z_{jq} = \langle T^{(j)}, T^{(q)} \rangle_F$ factors into the product of component inner products:

$$Z_{jq} = \prod_{i=1}^{p} \langle \tilde{v}_i^{(j)}, \tilde{v}_i^{(q)} \rangle \tag{21}$$

### B.2. Inner Product Statistics for Independent Contexts

Before analyzing the moments of $Z_{jq}$, we establish the key statistical property of inner products between vectors projected onto different random subspaces.

Let $P_1^{\perp}, P_2^{\perp}$ be projections onto independently generated random $(d-k)$-dimensional subspaces. Let $u, v \sim \mathcal{N}(0, I_d)$ be independent, and define:

$$\tilde{u} = \frac{P_1^{\perp} u}{\|P_1^{\perp} u\|}, \quad \tilde{v} = \frac{P_2^{\perp} v}{\|P_2^{\perp} v\|} \tag{22}$$

**Claim:** $\mathbb{E}[\langle \tilde{u}, \tilde{v} \rangle^2] = \frac{1}{d}$

*Proof.* Conditioned on the subspaces, $\tilde{u}$ and $\tilde{v}$ are independent uniform unit vectors in their respective $(d-k)$-dimensional subspaces. Let $E, F \in \mathbb{R}^{d \times (d-k)}$ be orthonormal bases for these subspaces. Then $\tilde{u} = E\alpha$ and $\tilde{v} = F\beta$ where $\alpha, \beta$ are uniform on $S^{d-k-1}$.

The inner product is:

$$\langle \tilde{u}, \tilde{v} \rangle = \alpha^T E^T F \beta = \alpha^T G \beta \tag{23}$$

where $G = E^T F$. For uniform vectors on $S^{d-k-1}$:

$$\mathbb{E}[\langle \tilde{u}, \tilde{v} \rangle^2 \mid G] = \frac{\|G\|_F^2}{(d-k)^2} \tag{24}$$

Since $P_2^\perp = FF^T$:

$$\|G\|_F^2 = \|E^T F\|_F^2 = \text{tr}(E^T FF^T E) = \text{tr}(E^T P_2^\perp E) \tag{25}$$

Taking expectations over independent random subspaces, $\mathbb{E}[P_2^\perp] = \frac{d-k}{d} I_d$, so:

$$\mathbb{E}[\|G\|_F^2] = \frac{d-k}{d}\text{tr}(E^T E) = \frac{(d-k)^2}{d} \tag{26}$$

Therefore:

$$\mathbb{E}[\langle \tilde{u}, \tilde{v}\rangle^2] = \frac{(d-k)^2/d}{(d-k)^2} = \frac{1}{d} \tag{27}$$

$\square$

### B.3. Moment Analysis

We now analyze the moments of $Z_{jq}$ using the result above.

**1. Signal Term ($j = q$).** In the case of a match, we are taking the inner product of identical unit vectors.

$$\langle \tilde{v}_i^{(q)}, \tilde{v}_i^{(q)}\rangle = 1 \implies Z_{qq} = \prod_{i=1}^{p}(1) = 1 \tag{28}$$

Thus, the signal is deterministic and so

$$\mathbb{E}[Z_{qq}] = 1, \quad \text{Var}(Z_{qq}) = 0 \tag{29}$$

**2. Interference Term ($j \neq q$).** In the case of a mismatch, $\tilde{v}_i^{(j)}$ and $\tilde{v}_i^{(q)}$ are derived from independent Gaussian vectors projected onto independently generated $(d-k)$-dimensional subspaces. By the claim above:

$$\mathbb{E}[\langle \tilde{v}_i^{(j)}, \tilde{v}_i^{(q)}\rangle] = 0, \quad \mathbb{E}[\langle \tilde{v}_i^{(j)}, \tilde{v}_i^{(q)}\rangle^2] = \frac{1}{d} \tag{30}$$

Since the components $i = 1 \ldots p$ are independent, the expectation of the product is the product of the expectations:

$$\mathbb{E}[Z_{jq}] = \prod_{i=1}^{p} \mathbb{E}[\langle \tilde{v}_i^{(j)}, \tilde{v}_i^{(q)}\rangle] = 0 \tag{31}$$

The variance calculation follows:

$$\text{Var}(Z_{jq}) = \mathbb{E}[Z_{jq}^2] = \prod_{i=1}^{p} \mathbb{E}[\langle \tilde{v}_i^{(j)}, \tilde{v}_i^{(q)}\rangle^2] = \left(\frac{1}{d}\right)^p = \frac{1}{d^p} \tag{32}$$

### B.4. Total Retrieval Variance

The retrieval score is $S = \sum_{j=1}^{N} Z_{jq} = Z_{qq} + \sum_{j \neq q} Z_{jq} = 1 + I$, where $I = \sum_{j \neq q} Z_{jq}$ is the interference. Since the interference terms are independent with zero mean and variance $\frac{1}{d^p}$:

$$\mathbb{E}[S] = 1, \quad \text{Var}(S) = \text{Var}(I) = \frac{N-1}{d^p} \tag{33}$$

# C. Clifford Generalization of TPR

Chisolm (2012) provides the needed background on Clifford algebra. We conclude this section by providing motivation for the rejection operator.

## C.1. Motivation

The carving operator $P_{\mathcal{C}}^{\perp}$ from Definition 4.3 can be expressed in Clifford algebra without matrices. The idea is simple: instead of representing the context subspace with a basis matrix, we encode it as a single object called a blade.

Let $\mathcal{C}$ be a context with orthogonal vectors $r_1, \ldots, r_k$. Their wedge product $A_r = r_1 \wedge \cdots \wedge r_k$ is a $k$-blade representing the subspace they span. This blade *is* the context, not a matrix describing it, but the subspace itself as an algebraic object.

Clifford algebra provides a natural decomposition of any vector $a$ into components parallel and perpendicular to this subspace:

$$P_{A_r}(a) + R_{A_r}(a) = a, \quad \text{where} \quad P_{A_r}(a) = (a \,\lrcorner\, A_r)A_r^{-1}.$$

The projection $P_{A_r}(a)$ lies inside the context; the rejection $R_{A_r}(a)$ is what remains after removing that component. This rejection is exactly the carving operation - it strips away whatever part of $a$ overlaps with the forbidden subspace.

In this formulation, fillers also become subspaces. A filler is now a $p$-blade $F = v_1 \wedge \cdots \wedge v_p$ rather than a tuple of vectors. To bind these two subspaces, we reject each component individually and wedge the results:

$$R_{A_r}(v_1 \wedge \cdots \wedge v_p) := R_{A_r}(v_1) \wedge \cdots \wedge R_{A_r}(v_p).$$

The output is a blade lying entirely in the orthogonal complement of the context, which is precisely what OSC binding requires. This perspective offers geometric clarity, but proving properties in this setting is more involved, so we use the GPU-friendly linear algebra formulation throughout the paper.

OSC is not alone in admitting a Clifford-algebraic formulation. We generalize TPR using the framework of Clifford algebras. Let $\mathrm{Cl}(n)$ denote the Clifford algebra generated by $n$ basis vectors squaring to 1. We show that representing roles and fillers as vectors in this algebra yields a strict generalization of classical TPR.

## C.2. Recursive Unbinding Theorem

Most proofs are Lean-verified.

**Definition C.1** (Vector Space Assumptions). Let $V$ be a real inner product space with a symmetric, bilinear inner product denoted by $(\cdot)$. We assume the standard left contraction (interior product) on the exterior algebra $\bigwedge V$.

**Definition C.2** (Contraction Convention). We adopt the standard convention that left contraction is left-associative and evaluated sequentially from the inside out:

$$a_r \lrcorner \ldots \lrcorner a_1 \lrcorner B \equiv a_r \lrcorner (\ldots (a_1 \lrcorner B) \ldots)$$

**Lemma C.3** (Iterated Contraction Identity). *Let $U = (u_1, \ldots, u_r)$ be a sequence of $r$ query vectors and $V = (v_1, \ldots, v_{r+1})$ be a sequence of $r + 1$ target vectors forming a blade $B = v_1 \wedge \cdots \wedge v_{r+1}$. The iterated contraction of $U$ onto $B$ is given by:*

$$u_r \lrcorner (\ldots (u_1 \lrcorner B) \ldots) = \sum_{\sigma \in S_{r+1}} \mathrm{sgn}(\sigma) \left( \prod_{\ell=1}^{r} u_\ell \cdot v_{\sigma(\ell)} \right) v_{\sigma(r+1)}$$

*Proof.* This is a standard result in exterior algebra relating iterated contractions to the generalized Laplace expansion of a determinant (Doran & Lasenby, 2003).

While the contraction of a single vector introduces a position-dependent sign $(-1)^{m-1}$, the cumulative sign for a sequence of $r$ contractions corresponds exactly to the signature of the permutation $\sigma$. Specifically, there is a natural bijection between valid contraction paths (sequences of distinct indices removed from $B$) and the symmetric group $S_{r+1}$. The permutation $\sigma \in S_{r+1}$ is defined by mapping the contraction step $\ell$ to the target index $\sigma(\ell)$, and mapping $r + 1$ to the survivor index $\sigma(r + 1)$. The sign $(-1)^{m-1}$ for removing the $m$-th vector accumulates multiplicatively over $r$ contractions, resulting in $\mathrm{sgn}(\sigma)$ due to the parity of the permutation's inversion count. $\qquad \square$

**Theorem C.4** (Recursive Unbinding). *Let $r \geq 1$. Consider a bundled object $A$ consisting of $p$ terms:*

$$A = \sum_{j=1}^{p} a_{j_1} \wedge a_{j_2} \wedge \cdots \wedge a_{j_r} \wedge f_j$$

***Notation:*** *Let $u = (u_1, \ldots, u_r) = (a_{i_1}, \ldots, a_{i_r})$ be the sequence of query vectors. For each term $j$, let $v^{(j)} = (a_{j_1}, \ldots, a_{j_r}, f_j)$ be the sequence of target vectors.*

*The result of recursively unbinding $A$ using the query $u$ is:*

$$a_{i_r} \lrcorner (\cdots (a_{i_1} \lrcorner A) \cdots) = \sum_{j=1}^{p} \left( d_j f_j + \sum_{k=1}^{r} c_{j_k} a_{j_k} \right)$$

*where the coefficients are defined by sums over the symmetric group $S_{r+1}$:*

$$d_j = \sum_{\sigma \in G_{r+1}} (\mathrm{sgn}\, \sigma) \prod_{\ell=1}^{r} (u_\ell \cdot v^{(j)}_{\sigma(\ell)})$$

$$c_{j_k} = \sum_{\sigma \in (k, r+1) G_{r+1}} (\mathrm{sgn}\, \sigma) \prod_{\ell=1}^{r} (u_\ell \cdot v^{(j)}_{\sigma(\ell)})$$

*Here:*

- $G_{r+1} = \mathrm{Stab}(r+1) \cong S_r$ *is the subgroup of permutations fixing index $r+1$.*

- $(k, r+1)$ *denotes the transposition swapping indices $k$ and $r+1$.*

- $(k, r+1) G_{r+1}$ *is the left coset consisting of all $\sigma$ such that $\sigma(r+1) = k$.*

*Remark C.5. If $r = 0$, the contraction trivially returns $A$, consistent with the empty product in the coefficient definitions.*

C.2.2. PROOF

*Proof.* By linearity of the contraction and the dot product, it suffices to prove the result for a single term $B = v_1 \wedge \cdots \wedge v_{r+1}$. For clarity, in the proof we drop the superscript $(j)$ when analyzing a single term, letting $v = (a_{j_1}, \ldots, a_{j_r}, f_j)$.

**Step 1: Application of the Contraction Lemma.** We apply Lemma C.3 directly to the term $B$. The iterated contraction yields a summation over the symmetric group $S_{r+1}$:

$$\text{Result} = \sum_{\sigma \in S_{r+1}} (\mathrm{sgn}\, \sigma) \left( \prod_{\ell=1}^{r} u_\ell \cdot v_{\sigma(\ell)} \right) v_{\sigma(r+1)}$$

This step effectively expands the determinants of the interaction matrices, grouping terms by which vector $v_{\sigma(r+1)}$ survives the contraction process.

**Step 2: Partitioning the Symmetric Group.** We partition the sum based on the index of the surviving vector, determined by $\sigma(r+1)$. The set of indices is $\{1, \ldots, r+1\}$.

**Case A: The filler survives ($\sigma(r+1) = r+1$).**
The subset of permutations satisfying this condition is exactly the stabilizer subgroup $G_{r+1}$. For these terms, the surviving vector is $v_{r+1} = f_j$. The contribution to the sum is:

$$\left( \sum_{\sigma \in G_{r+1}} (\mathrm{sgn}\, \sigma) \prod_{\ell=1}^{r} (u_\ell \cdot v_{\sigma(\ell)}) \right) f_j$$

This matches the definition of $d_j f_j$.

**Case B: A role survives ($\sigma(r+1) = k$ for some $k \in \{1, \ldots, r\}$).**
For a fixed $k$, the set of permutations mapping $r + 1$ to $k$ corresponds to the left coset formed by composing the stabilizer with the transposition $\tau = (k, r+1)$:

$$\{\sigma \in S_{r+1} \mid \sigma(r+1) = k\} = (k, r+1)G_{r+1}$$

This bijection ensures the coset $(k, r+1)G_{r+1}$ exhaustively enumerates all permutations with $\sigma(r+1) = k$, as left multiplication by $\tau$ is invertible.

For these terms, the surviving vector is $v_k = a_{j_k}$. The contribution is:

$$\sum_{k=1}^{r} \left( \sum_{\sigma \in (k,r+1)G_{r+1}} (\operatorname{sgn} \sigma) \prod_{\ell=1}^{r} (u_\ell \cdot v_{\sigma(\ell)}) \right) a_{j_k}$$

This matches the definition of $\sum c_{j_k} a_{j_k}$.

**Conclusion.**  Summing the results from Case A and Case B yields the expression in the theorem statement.  $\square$

### C.3. Exact Recovery

**Theorem C.6** (Recursive Clifford Extension of TPR). *Let $a_1, \ldots, a_r$ be vectors in $\operatorname{Cl}(n+m)$ with support only on the first $n$ basis vectors, and let $f_1, \ldots, f_p$ be vectors with support only on the last $m$ basis vectors. Assume the roles $a_i$ are orthonormal. For a bundled object $A = \sum_{j=1}^{p} a_{j_1} \wedge a_{j_2} \wedge \cdots \wedge a_{j_r} \wedge f_j$, assume that the roles in each term are written in a canonical order (e.g., $j_1 < j_2 < \cdots < j_r$). Then unbinding with the matching sequence of roles recovers the filler exactly:*

$$a_{i_r} \lrcorner \left( a_{i_{r-1}} \lrcorner \left( \cdots (a_{i_1} \lrcorner A) \cdots \right) \right) = f_i$$

*where $i$ is the unique index such that $(a_{i_1}, \ldots, a_{i_r}) = (a_{j_1}, \ldots, a_{j_r})$, and the result is $0$ if no such $j$ exists.*

*Proof.* By the Recursive Unbinding theorem, with $(v_1, \ldots, v_{r+1}) = (a_{j_1}, \ldots, a_{j_r}, f_j)$, we have

$$a_{i_r} \lrcorner (\cdots (a_{i_1} \lrcorner A) \cdots) = \sum_{j=1}^{p} \left( d_j f_j + \sum_{k=1}^{r} c_{j_k} a_{j_k} \right)$$

where

$$d_j = \sum_{\sigma \in G_{r+1}} (\operatorname{sgn} \sigma) \prod_{\ell=1}^{r} (a_{i_\ell} \cdot v_{\sigma(\ell)})$$

$$c_{j_k} = \sum_{\sigma \in (k,r+1)G_{r+1}} (\operatorname{sgn} \sigma) \prod_{\ell=1}^{r} (a_{i_\ell} \cdot v_{\sigma(\ell)})$$

For $d_j$: since $\sigma \in G_{r+1}$ fixes $r + 1$, we have $\sigma(\ell) \in \{1, \ldots, r\}$ for all $\ell \le r$, so each $v_{\sigma(\ell)} = a_{j_{\sigma(\ell)}}$ is a role. Thus

$$d_j = \sum_{\sigma \in G_{r+1}} (\operatorname{sgn} \sigma) \prod_{\ell=1}^{r} (a_{i_\ell} \cdot a_{j_{\sigma(\ell)}})$$

By orthonormality, $a_{i_\ell} \cdot a_{j_{\sigma(\ell)}} = \delta_{i_\ell, j_{\sigma(\ell)}}$. The product is nonzero only if $i_\ell = j_{\sigma(\ell)}$ for all $\ell$. Since both the query sequence $(i_1, \ldots, i_r)$ and the stored sequence $(j_1, \ldots, j_r)$ are in canonical order, the only permutation that can satisfy this is the identity. Thus $d_j = 1$ if $(i_1, \ldots, i_r) = (j_1, \ldots, j_r)$, and $d_j = 0$ otherwise.

For $c_{j_k}$: since $\sigma \in (k, r+1)G_{r+1}$ sends $r + 1$ to $k$, there exists some $\ell$ with $\sigma(\ell) = r + 1$, meaning $v_{\sigma(\ell)} = f_j$. Since roles and fillers have disjoint support, $a_{i_\ell} \cdot f_j = 0$, so the entire product vanishes. Thus $c_{j_k} = 0$ for all $j, k$.

Therefore

$$a_{i_r} \lrcorner (\cdots (a_{i_1} \lrcorner A) \cdots) = \sum_{j=1}^{p} d_j f_j = f_i$$

where $i$ is the index with $(a_{i_1}, \ldots, a_{i_r}) = (a_{j_1}, \ldots, a_{j_r})$, and the result is 0 if no such $j$ exists. $\square$

### C.4. Compressing TPR via Orthogonal Complements

The proofs above only use the disjoint support of roles and fillers to establish orthogonality. This suggests a relaxation: if we instead require that fillers live in the orthogonal complement of the roles, exact recovery still holds, even when roles and fillers share the same basis vectors.

This observation has implications for parameter efficiency. Letting roles and fillers live on completely disjoint supports requires $n^{|\mathcal{C}|+1}$ parameters (as it is TPR). Letting them share all basis vectors while enforcing orthogonality reduces this to $\binom{n}{|\mathcal{C}|+1}$ parameters. In other words, we can compress the TPR representation while keeping it lossless by constraining fillers to the orthogonal complement of the role subspace—precisely the geometric relationship that OSC enforces through subspace carving.

# D. Experiments

## D.1. Extended Experimental Results

Prior to presenting more experimental results, we outline key implementation details. Context generation requires QR decomposition of a $d \times k$ matrix costing $\mathcal{O}(dk^2)$ operations, independent of context size. Simple testing shows that $k$ close to $d - \sqrt{d}$ is optimal. Therefore, instead of computing the carving operator through the projection matrix, we compute the operator directly, generating only $k_1 = d - k$ orthogonal vectors. This is negligible for $k_1 \ll d$ and extremely parallelizable across contexts (simply outer products and additions). Frequently accessed contexts can be cached.

**Other baseline VSA experiments.** We give more experiments. Figure 6 shows OSC's performance against VSAs of dimension $d = 8100$. To showcase the fact that OSC is invariant to context size, we also perform the same experiment for role depth of 48 and compare against VSAs with dimension $d = 8100$ in Figure 7. This dimension was chosen as it is a perfect square, which required by VTB, that is close to 8192. Figure 8 shows OSC's performance with $p = 2$ against VSA when superposition memory is held constant. As with Figure 5, OSC beats the VSA with the highest dimension with fewer params that the VSA with the lowest dimension. We presented retrieval accuracy for OSC with $p = 3$ in Figure 5 in §6; we present both the retrieval and recognition accuracy in Figure 9.

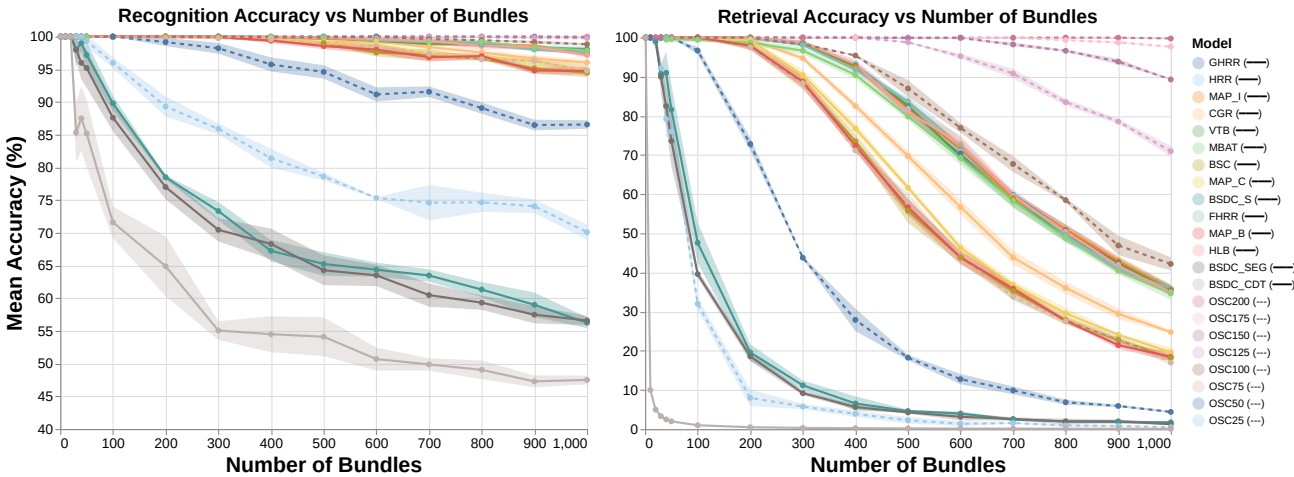

*Figure 6.* Performance comparison of 14 VSAs at $d = 8100$ and OSC across $d = 25, 50, 75, 100, 125, 150, 175,$ and $200$ for $p = 2$ at increasing bundle counts with unique fillers and role depth 1. OSC outperforms all VSAs on both retrieval and recognition tasks. Note the difference in $Y$ axes between graphs. Standard deviation is shown as the shaded regions.

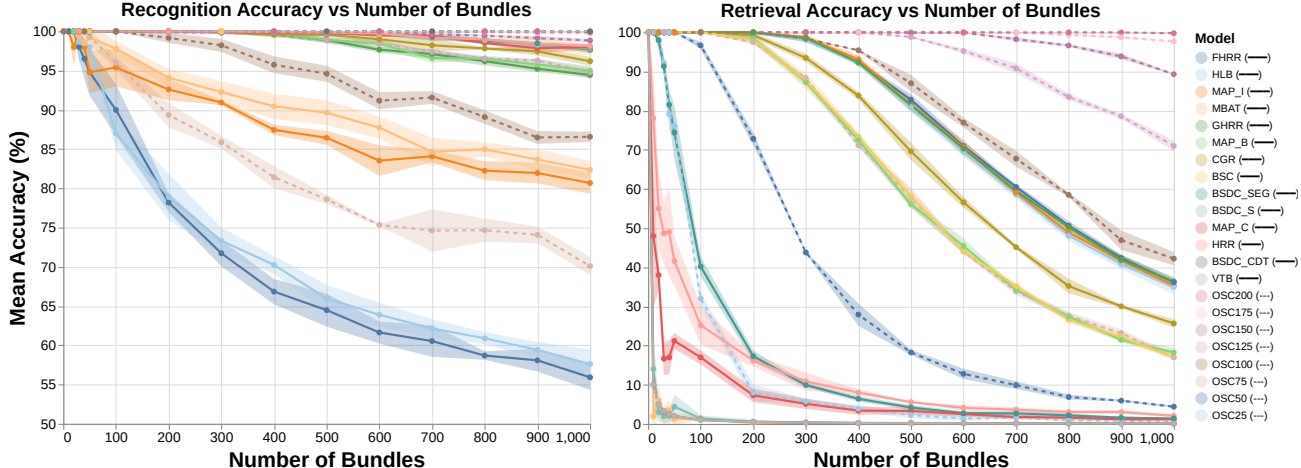

*Figure 7.* Performance comparison of 14 VSAs at $d = 8100$ and OSC across $d = 25, 50, 75, 100, 125, 150, 175,$ and $200$ for $p = 2$ at increasing bundle counts with unique fillers and role depth 48. OSC outperforms all VSAs on both retrieval and recognition tasks. Note the difference in $Y$ axes between graphs. Standard deviation is shown as the shaded regions. OSC's performance for a role depth of 48 is the exact same as for a depth of 1 (Figure 6).

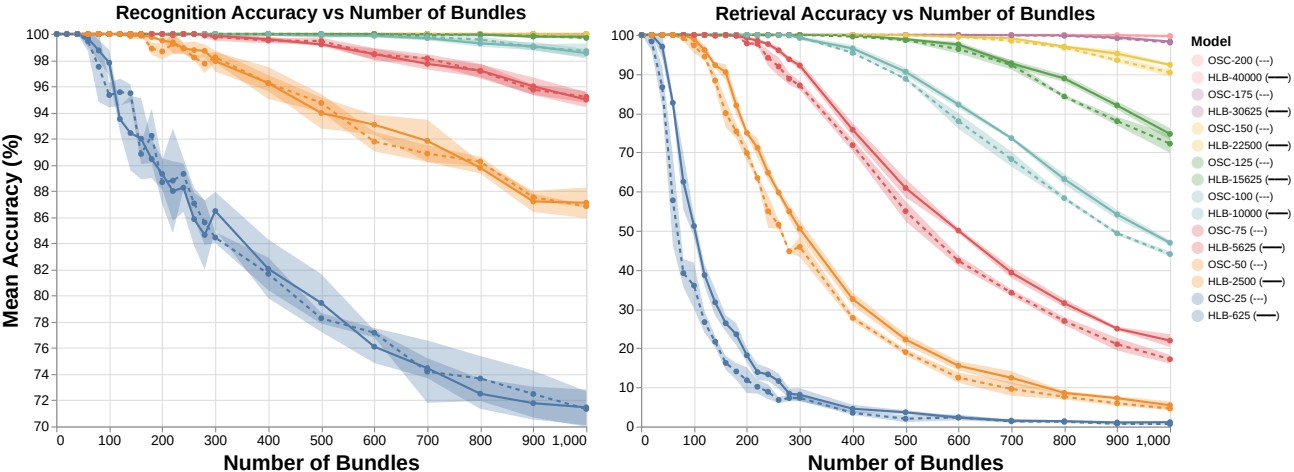

*Figure 8.* Performance of HLB at varying dimensions against OSC for $p = 2$. Identically colored runs share the same superposition memory size (not total memory). VSAs are slightly superior than OSC in memory capacity, but OSC's sub-linear scaling requires the VSA to have $200^2 * 1001 / (200^2 + 2 * 200 * 1000) = 91\times$ OSC's total memory to get close in retrieval accuracy at 1000 bundles. Standard deviations are denoted by the shaded regions.

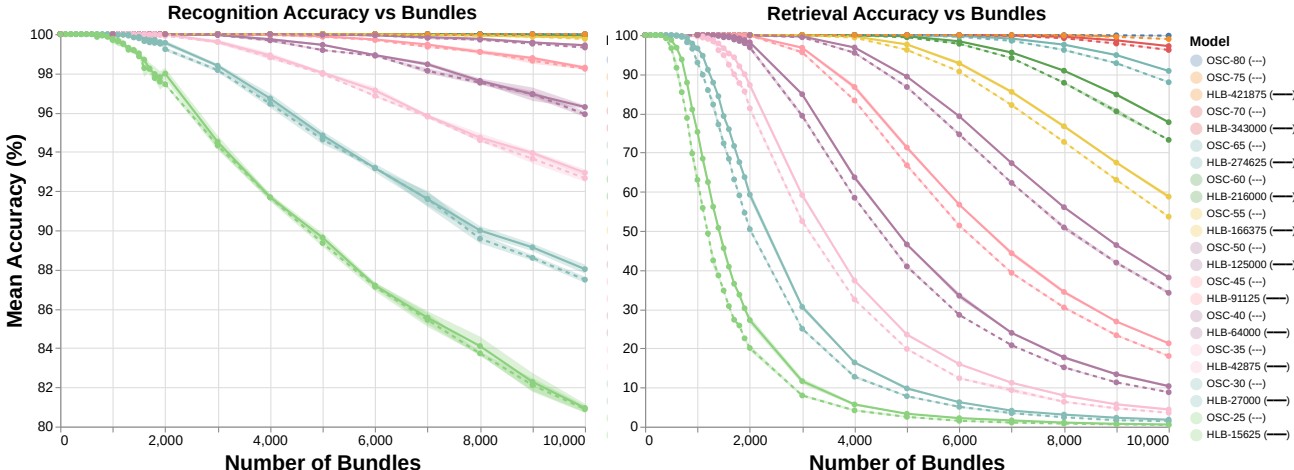

*Figure 9.* Performance of HLB at varying dimensions against OSC for $p = 3$. Identically colored runs share the same superposition memory size (not total memory). VSAs are slightly superior than OSC in memory capacity, but OSC's sub-linear scaling requires the VSA to have $70^3 * 10001 / \left( 80^3 + 3 * 80 * 10000 \right) \approx 1178 \times$ OSC's total memory to achieve comparable retrieval accuracy. Standard deviations (shaded regions) are present but negligible.

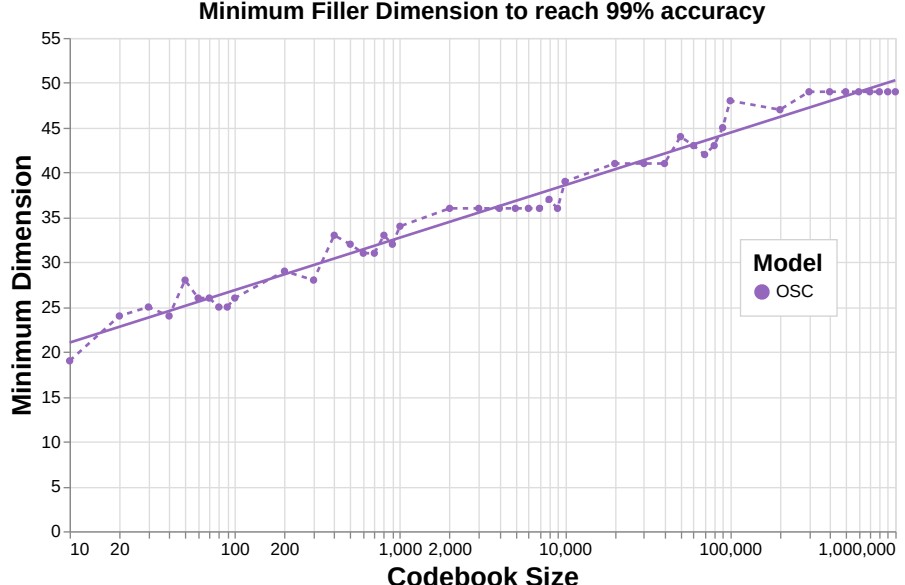

*Figure 10.* Adapted from Figure 4, this plot excludes VSAs to highlight the logarithmic relationship between the filler dimension and codebook size for OSC. Minimum filler dimension required to achieve 99% accuracy. A dimension was selected only if mean accuracy across 10 trials exceeded 99% at that dimension but fell below 99% at the next lowest dimension. OSC exhibits the same favorable logarithmic scaling with codebook size as VSAs. The dotted line tracks the actual dimension points, while the solid line represents the best fit.

## D.2. Discriminative Recognition vs Generative Retrieval

We evaluated OSC retrieval with recognition. To determine if this methodology impacts VSA performance, we applied the same recognition-based retrieval to FHRR, HLB, MAP_I, and VTB. For FHRR, HLB, and MAP_I, binding and unbinding are adjoint operations, so $\langle \text{Unbind}(M, r), g \rangle = \langle M, \text{bind}(r, g) \rangle$ for every candidate $g$, and the two methods return the same $\arg\max$. VTB does not satisfy this adjoint relation, but we observed no significant difference between standard retrieval and recognition-based retrieval at dimensions $d = 4096$ and $d = 8100$. As shown in Figure 11, the accuracy closely mirrors the results in Figures 3 and 6.

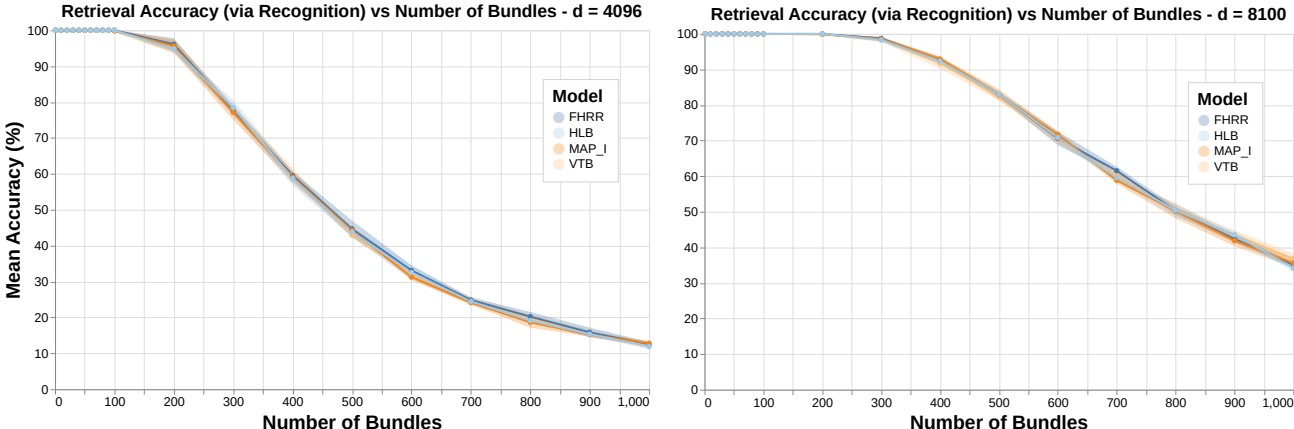

*Figure 11.* Retrieval accuracy when using the recognition-based retrieval approach for FHRR, HLB, MAP_I, and VTB. The curves for $d = 4096$ and $d = 8100$ match those of Figure 3 and Figure 6. Shaded regions denote standard deviations.

## D.3. Carving dimension.

We investigate the optimal carving dimension $k_1$ into which filler components are projected. The trade-off involves two competing terms, cross-context interference (favoring large $k_1$) and within-context filler discrimination (favoring small $k_1$). Empirically, we find that scaling the carving dimension with $\mathcal{O}(\sqrt{d})$ to be optimal. While further analysis is needed to determine if the true optimum scales as $\mathcal{O}\left(d^{\frac{1}{p}}\right)$, the practical distinction is negligible in our regime. The flexibility of the tensor order $p$ allows us to operate with relatively small filler dimensions, where a broad range of subspace sizes yield stable results.

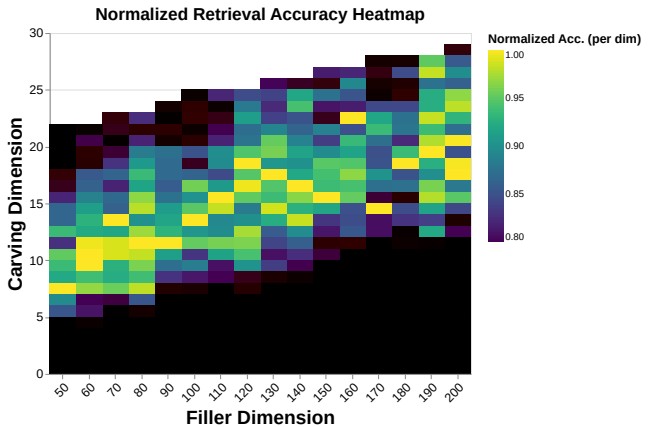

*Figure 12.* Heatmap of retrieval accuracy for filler dimensions ranging from 50 to 200 in increments of 10. Accuracy is normalized per dimension, with the maximum value scaled to 1.

### D.4. FLOP Analysis

We complement the wall-clock measurements in Table 3 with a FLOP analysis of the retrieval operation. Counts are derived from the matrix dimensions in the timed code.

**HLB.** Retrieval consists of an elementwise division to unbind the role and a single matrix-vector product against the candidate codebook. The dominant cost is

$$\text{FLOPs}_{\text{HLB}} \approx 2Ld, \tag{34}$$

where $L$ is the vocabulary size and $d$ is the dimension.

**OSC.** Retrieval consists of three steps. First, the complement role-subspace projector $P = r^\top r$ is formed, costing $2k_1 d^2$ where $k_1$ is the carving dimension. Second, the candidates are projected through $P$, costing $2Lpd^2$. Note that rejecting onto the orthogonal complement is the same as projecting onto the subspace. Third, the projected candidates are contracted against the order-$p$ memory tensor, costing $2Ld^p$. The dominant cost is

$$\text{FLOPs}_{OSC} \approx 2k_1 d^2 + 2Lpd^2 + 2Ld^p. \tag{35}$$

**Benchmark FLOP counts.** Table 5 reports total FLOPs for the three configurations of Table 3, using $k_1 = \lfloor \sqrt{d} \rfloor$ as in the timing code. **B1**: OSC ($d = 64, p = 2$) vs HLB ($d = 4{,}096$). **B2**: OSC ($d = 200, p = 2$) vs HLB ($d = 40{,}000$). **B3**: OSC ($d = 75, p = 3$) vs HLB ($d = 421{,}875$).

|  |  | $L = 100$ | $L = 1{,}000$ | $L = 10{,}000$ |
|---|---|---|---|---|
| **B1** | HLB | 0.82M | 8.19M | 81.9M |
|  | OSC | 2.52M | 24.6M | 246M |
| **B2** | HLB | 8.0M | 80M | 800M |
|  | OSC | 25.1M | 241M | 2.4B |
| **B3** | HLB | 84.4M | 844M | 8.44B |
|  | OSC | 87.8M | 878M | 8.78B |

*Table 5.* Total FLOPs for HLB vs. OSC retrieval at the configurations of Table 3. OSC's FLOP count exceeds HLB's at every configuration, yet wall-clock timing favors OSC at scale because the bottleneck is memory bandwidth rather than arithmetic throughput. When OSC has $p = 2$, the FLOPs count is $\approx 3\times$ HLB, while when OSC has $p = 3$ the FLOPs count is approximately equal due to the very large filler dimension.

# E. Hyperparameters

Table 6 contains the hyperparameters for the XML experiments. Figure 12 discusses the optimal carving dimension based on the filler dimension of OSC.

*Table 6.* Complete hyperparameter configuration. Expansion refers to the width multiplier for the second hidden layer (e.g., $512 \times 2 = 1024$ for Eurlex). Abbreviations: BS (Batch Size), Drop (Dropout Rate), Output Dim (Output dimension for HLB/VTB/MAP), OSC d (Filler dimension for OSC). Our implementation builds upon the codebase of Alam et al. (2024). We utilize the optimal hyperparameters reported in their work. When running their code, we could not recreate HLB's dominance. We contacted the authors, but could not resolve the issue.

| Dataset | Epochs | BS | Hidden | Expansion | Drop | Output Dim | OSC d |
|---|---|---|---|---|---|---|---|
| BIBTEX | 10 | 64 | 512 | 1 | 0.00 | 400 | 125 |
| MEDIAMILL | 10 | 64 | 512 | 1 | 0.00 | 400 | 125 |
| DELICIOUS | 10 | 64 | 512 | 1 | 0.25 | 400 | 125 |
| EURLEX-4K | 10 | 256 | 512 | 2 | 0.35 | 1,600 | 200 |
| EURLEX-4.3K | 25 | 256 | 512 | 1 | 0.00 | 400 | 100 |
| WIKI10-31K | 25 | 16 | 512 | 1 | 0.25 | 3,000 | 300 |
| DELICIOUS-200K | 25 | 8 | 1024 | 1 | 0.25 | 3,000 | 300 |
| AMAZON-13K | 25 | 1024 | 512 | 1 | 0.25 | 3,000 | 300 |

