# OpenReview forum: "Recursive Binding on a Budget: Subspace Carving in Order-$p$ Tensor Memories"
_ICML.cc/2026/Conference — ICML 2026 regular_

### Official Review · Reviewer_ZxPM · 2026-03-07

**Soundness:** 3
**Presentation:** 3
**Significance:** 2
**Originality:** 3
**Overall Recommendation:** 4
**Confidence:** 3

**Summary:**

The paper introduces `Orthogonal Subspace Carving (OSC), a memory architecture that uses projection to bind fillers to context-specific subspaces. By defining forbidden zones rather than explicit slots, OSC achieves massive memory savings for deep recursive structures compared to standard tensor product representations (TPRs) and vector symbolic architectures (VSAs). It replaces noisy superposition with geometric orthogonality, enabling high-fidelity retrieval even with small vector dimensions.

**Compliance With Llm Reviewing Policy:**

Affirmed.

**Final Justification:**

I appreciate the authors' efforts and will maintain my positive score of 4.

**Key Questions For Authors:**

1.  How does the inference speed (wall-clock time) compare to standard VSAs?
2.  Can OSC handle out-of-vocabulary fillers during inference?

**Limitations:**

See Weaknesses. The primary limitation is the inability to perform exact algebraic unbinding (reconstruction), which restricts the architecture to recognition-based tasks where a vocabulary is available. The computational overhead of generating orthogonal bases (e.g., via QR decomposition) compared to simpler VSA operations is another point to consider for high-throughput applications.

**Strengths And Weaknesses:**

Strengths
1. Novel Idea: The core concept of defining structure via subspace constraints (where not to put information) is clever and mathematically sound.
2. Efficiency: Decoupling the component dimension from memory capacity is a game-changer. The results show 9-18x better storage efficiency than baselines.
3. Solid Validation: It works not just on synthetic tasks but also beats SOTA methods on real-world Extreme Multi-label Learning benchmarks.

Weaknesses
1. OSC relies on checking a vocabulary list rather than direct reconstruction. This limits its use in generative tasks.
2. The paper focuses on memory savings but doesn't fully discuss the computational cost of the projection operations (like QR decomposition) compared to simpler VSA ops.
3.  More guidance on choosing the tensor order $p$ for different tasks would be helpful.

---

> ### Author Rebuttal · Authors · 2026-03-31
>
> We are grateful for the reviewer's careful reading and insightful feedback. We answer your questions below.
>
> **- How does the inference speed (wall-clock time) compare to standard VSAs?**
>
> We initially omitted inference timings because each VSA has distinct binding/unbinding mechanisms, and the effort invested in low-level tuning would inevitably be uneven across 14 baselines, making comparisons potentially misleading.
>
> However, we appreciate the reviewer's interest in wall-clock performance and benchmark against HLB, the best-performing VSA from Figure 3, and will include a more comprehensive graph in the draft. HLB is one of the simplest VSAs with its binding and unbinding operations being elementwise multiplication and division.
>
> Below, we report retrieval timings with superposition memory held constant and amortized generation costs excluded. Because HLB's elementwise operations are simple, it is faster at small enough dimensions. However, as the representation scales through higher superposition, HLB's memory footprint dominates. At that point, even though OSC's per-element operations are more expensive, OSC becomes faster because the bottleneck is memory bandwidth, not compute. These were run on an A100 with $40$ GB of HBM. N represents the amount of superposition.
>
> Along with the speed, we have included the parameter count to illustrate the memory bandwidth problem VSAs incur at scale. A ratio $>1$ in the last row indicates that OSC is faster or uses less memory.
>
> **Benchmark 1: OSC: d=64, p=2 vs HLB: d=4,096**
> |  | Speed (ms) | | | Parameter Count | | |
> |:---|:---:|:---:|:---:|:---:|:---:|:---:|
> |  | N = 100 | N = 1,000 | N = 10,000 | N = 100 | N = 1,000 | N = 10,000 |
> | **HLB** | 0.068 | 0.071 | 0.180 | 414K | 4.1M | 41.0M |
> | **OSC** | 0.149 | 0.140 | 0.156 | 17K | 132K | 1.3M |
> | **HLB/OSC** | 0.46× | 0.50× | 1.15× | 24.5× | 31.0× | 31.9× |
>
> **Benchmark 2: OSC: d=200, p=2 vs HLB: d=40,000**
> |  | Speed (ms) | | | Parameter Count | | |
> |:---|:---:|:---:|:---:|:---:|:---:|:---:|
> |  | N = 100 | N = 1,000 | N = 10,000 | N = 100 | N = 1,000 | N = 10,000 |
> | **HLB** | 0.071 | 0.196 | 1.189 | 4.0M | 40.0M | 400.0M |
> | **OSC** | 0.155 | 0.203 | 0.306 | 80K | 440K | 4.0M |
> | **HLB/OSC** | 0.46× | 0.97× | 3.88× | 50.5× | 91.0× | 99.0× |
>
> **Benchmark 3: OSC: d=75, p=3 vs HLB: d=421,875**
> |  | Speed (ms) | | | Parameter Count | | |
> |:---|:---:|:---:|:---:|:---:|:---:|:---:|
> |  | N = 100 | N = 1,000 | N = 10,000 | N = 100 | N = 1,000 | N = 10,000 |
> | **HLB** | 0.261 | 1.345 | OOM | 42.6M | 422.3M | OOM |
> | **OSC** | 0.384 | 0.407 | 1.252 | 444K | 647K | 2.7M |
> | **HLB/OSC** | 0.68× | 3.31× | NA | 95.9× | 652.8× | NA |
>
>
> **- Can OSC handle out-of-vocabulary fillers during inference?**
>
> In its current formulation, OSC relies on scoring against a fixed codebook, so it does not natively handle out-of-vocabulary fillers. However, this limitation is shared by any VSA or TPR system that uses a cleanup memory for retrieval. Whether OSC can support out-of-vocabulary fillers in a learned setting, as architectures like sDTM do through content-structure factorization, is an open question that we plan to explore in future work.
>
> **- More guidance on choosing the tensor order $p$ for different tasks would be helpful.**
>
> Providing an optimal $p$ per task is difficult as it depends on the specific superposition, vocabulary, accuracy, and speed requirements. While increasing $p$ leads to a smaller filler dimension for the same accuracy, this requires the vocabulary to be large enough for the memory benefits to be realized. In our experiments, we found $p=2,3$ to be sufficient.

---

> > ### Author Rebuttal · Reviewer_ZxPM · 2026-04-02
> >
> > Thank you to the authors for their responses.  All problems are resolved

---

### Official Review · Reviewer_B7gL · 2026-03-08

**Soundness:** 2
**Presentation:** 3
**Significance:** 2
**Originality:** 2
**Overall Recommendation:** 3
**Confidence:** 2

**Summary:**

The contribution of this paper is a symbolic memory architecture called Orthogonal Subspace Carving (OSC) which is positioned between tensor product representations and vector symbolic architectures. The key idea is, rather than assigning bindings to explicit coordinate slots, to define instead where infromation is *not* allowed to go by associating a "forbidden" subspace; bindings are placed in the orthogonal complement of this subspace. Orthogonality is required only within a single context; different contexts need not share any geometric relationshp, providing structural precision within each context but independence across contexts. The goal is recast to support recognition (testing whether a specific binding is present) rather than recall (reconstructing content from a cue).

For transparency, this topic is somewhat outside my area of expertise, particularly the VSA literature.

**Compliance With Llm Reviewing Policy:**

Affirmed.

**Key Questions For Authors:**

- How should the reader interpret the claim that OSC “approximates the lossless-ness of TPRs”?
- Can the authors provide a more complete computational complexity analysis?
- Can the authors discuss the relation to key-value neural memory and attention-style retrieval? OSC seems closer to recognition-by-scoring than to algebraic unbinding, suggesting a natural comparison to attention and modern associative memory mechanisms. Would an empirical comparison be relevant?

**Strengths And Weaknesses:**

Strengths
- Presentation is strong. I appreciated the preliminaries section, which helped position the paper's contributions. Throughout, the paper's motivation and claims are laid out very clearly. E.g., the framing of "the memory triangle" is very clear.
- The construction of OSC is clean and mathematically simple. The core idea of defining "forbidden" spaces, where fillers are projected onto the orthogonal complement, is technically interesting.
- The interference analysis in Theorem 4.6 provides a concise and formal intuition, even if idealized, giving the method stronger analytical support.

Weaknesses
- At times, there is a mismatch between the papers strongest claims and what the method actually supports. For example, the introduction claims "a binding capability the approximates the lossless-ness of TPRs", but OSC explicitly gives up recall and exact unbinding so the the guarantee is not TPR-like lossless reconstruction, but recognition under a candidate set.
- The proposed method is favorable only when redefining the task from recall to recongition. The paper argues that "recognition is unavoidable" because cleanup memories already perform vocabulary search. Although this is an interesting observation, I did not find this fully persuasive. It seems to me that replacing unbinding with exhaustive cnadidate scoring changes the computational profile and the semantics of retrieval.
- The focus is on parameter complexity and storage efficiency, but the computational complexity is not addressed sufficiently. It seems the query-time cost would be substational. This should be quantified and reported in the main paper.
- Experiments are somewhat narrow relative to the paper's main claims. They support the method in synthetic retrieval settings, but do not establish how broad or practically meanigful the claimed advantages are beyond that synthetic domain.
- Although the related work section briefly cites work on dense associative memory and notes its connection modern deep learning (namely, attention), it does not sufficiently discuss how that line of work relates to OSC in terms of positioning. This is particularly relevant because those models are also motivated by higher-order interactions and storage capacity, which seems closely related to OSC’s framing around storage efficiency. A deeper discussion of why OSC is needed relative to modern neural memory alternatives would strengthen the paper’s positioning.

---

> ### Author Rebuttal · Authors · 2026-03-31
>
> We thank the reviewer for their detailed and constructive feedback. We address each question below.
>
> **- The paper argues that "recognition is unavoidable" because cleanup memories already perform vocabulary search. Although this is an interesting observation, I did not find this fully persuasive. It seems to me that replacing unbinding with exhaustive candidate scoring changes the computational profile and semantics of retrieval.**
>
> We agree that replacing unbinding with candidate scoring appears to fundamentally change how VSAs operate. However, for VSAs where the unbinding operation is the adjoint of binding, which includes classical architectures such as HRR, FHRR, and MAP, as well as the more recent HLB, retrieval via recognition yields mathematically identical rankings to standard unbinding-based retrieval, since the adjoint property guarantees the argmax over the codebook is equal. This property does not hold for VTB, but we observe no significant difference from standard retrieval as shown in Appendix D.2. We acknowledge that this connection should have been made explicit, and have added it to the revised paper. We discuss the favorable computational profile of OSC at scale in the next response.
>
> **- Can the authors provide a more complete computational complexity analysis?**
>
> Thank you for the great question! We initially omitted inference timings because fairly benchmarking 14 baselines with distinct binding primitives would require comparable optimization effort for each, and uneven tuning would make the comparison misleading. However, we realize that wall-clock time is important and have provided timings against HLB, one of the simplest VSAs with operations being elementwise multiplication and division. Because of the large memory savings, even though OSC operations are more expensive than VSAs, there is a speed-up at scale because the bottleneck is no longer compute-bound, but memory-bound. We provide a more detailed answer with wall-clock timings in our answer to Reviewer ZxPM where we show that OSC becomes faster than HLB as the representation scales, and that HLB runs out of memory in regimes where OSC remains tractable.
>
> **- Experiments are somewhat narrow relative to the paper's main claims. They support the method in synthetic retrieval settings, but do not establish how broad or practically meaningful the claimed advantages are beyond that synthetic domain.**
>
> We validate OSC's learnability on eight XML classification datasets where it achieves competitive accuracy against VSAs while using substantially fewer parameters (see our response to Reviewer PRJo for parameter Table). Broader evaluation, such as integrating OSC into architectures like sDTM as suggested by Reviewer Vozi, is a substantial effort that warrants its own dedicated study, one that the architecture and codebase established in this work support.
>
> **- How should the reader interpret the claim that OSC approximates the lossless-ness of TPRs?**
>
> We intended to convey that one can allocate a larger superposition memory while still achieving massive total memory savings over VSAs because OSC's filler dimension scales sub-linearly with memory capacity. This allows OSC to operate in a regime closer to the zero-interference guarantee of TPRs, without incurring their exponential dimensionality cost. We acknowledge that the original phrasing could be misread as claiming an algebraic approximation to TPR unbinding, which is not the case. We have revised the statement in the draft to: "near-lossless retrieval accuracy in high-superposition regimes, enabled by sub-linear filler scaling that allows for larger superposition memory spaces within a smaller total memory footprint without TPR's exponential dimensionality cost."
>
> **- Can the authors discuss the relation to key-value neural memory and attention-style retrieval? Would an empirical comparison be relevant?**
>
> We agree that the related work should better position OSC relative to dense associative memories and attention-style retrieval, and will expand this discussion in the final draft. The reviewer correctly observes that OSC's recognition-by-scoring resembles attention, and that at the level of a single binding, key-value retrieval and role-filler association are analogous.
>
> However, the underlying memory models are fundamentally different. Key-value neural memories maintain an explicit table of key-value pairs, where each binding occupies its own slot. Adding a new pair grows the memory size whereas OSC, VSAs, and TPR superpose all bindings into a single fixed-size trace.
>
> Additionally, attention's key-value store is inherently flat: each pair is independent with no native mechanism for expressing that one binding is nested inside another. VSA-style binding supports recursive compositional structure within a single trace. These differences make a direct empirical comparison difficult as the two paradigms operate under fundamentally different storage assumptions.

---

> > ### Author Rebuttal · Reviewer_B7gL · 2026-04-02
> >
> > Thank you to the authors for their responses. Their promised revisions help to improve the framing of the paper, and partially (but incompletely) address some of my main technical concerns. I have decided to maintain my current score, while also maintaining my low level of confidence as I am only vaguely familiar with the relevant literature.
> >
> > Please see below for point-by-point comments on the authors responses.
> >
> > * The rebuttal gives a more precisely stated claim than the initial version of the paper does, and this helps to improve the framing of the paper.
> > * I still find the redefinition of the problem to be a significant aspect, which I am unsure how to evaluate due to my limited familiarity with this literature. The fact that the method is incapable of recall due to the singularity, and requiring iteration through the vocabulary seems to be a major redefinition of the problem. I am unsure of the soundness or meaningfulness of 1) recasting the problem in this way, and 2) if we did, what should be the appropriate evaluation/comparison. I will defer the judgment of this to those with expertise in this area.
> > * The wall clock time numbers are appreciated and helpful. However, I think that a more thorough and careful analysis is needed for the paper. e.g., computational complexity, flops, and ensuring optimized implementations when reporting wall clock numbers, which is difficult to gauge from the numbers quoted in the rebuttals. Also, the speedup only occurs at very high superposition and very large representation settings. A discussion is needed to establish whether or not these are  the practically relevant regimes.
> > * I appreciate the authors acknowledgment and discussion of the relationship key-value neural attention mechanisms. I think that the paper would benefit significantly from addressing this head on and making the case for why VSA-style approaches are still important and viable given the empirical success of key-value neural attention mechanisms and OSC's use of recognition-by-scoring.

---

> > > ### Author Response · Authors · 2026-04-03
> > >
> > > We sincerely thank the reviewer for taking the time to respond to each one of our points. We respond below to your remaining concerns.
> > >
> > > **Bullet 2**
> > >
> > > The concern is whether recasting retrieval as recognition is a major redefinition of the problem. We view it is a simplification (not redefinition) because existing VSA retrieval already terminates with vocabulary search.
> > >
> > > Consider the end goal of VSA retrieval and the standard workflow: (1) Unbind the memory with the role to recover a noisy filler estimate. (2) Compare against every vocabulary entry using cosine similarity, and (3) return the highest-scoring entry. Step (2), known as cleanup memory, is unavoidable: even at moderate superposition, the noisy filler has very low similarity (below 0.1) to the true filler. Retrieval succeeds only because this weak signal exceeds the similarity with incorrect fillers, meaning **VSAs still iterate through the vocabulary**.
> > >
> > > If the end goal is to identify the correct filler, the question becomes: why must we unbind at all? The traditional workflow treats unbinding as necessary, but the final answer uses vocabulary search regardless. OSC takes advantage of this insight: rather than unbinding and then searching, it scores candidates directly against the memory. Both iterate through the vocabulary and return the best match; OSC's design yields less total memory for the same outcome.
> > >
> > > **Bullet 3**
> > >
> > > We agree that ensuring equally optimized implementations is difficult, which is why we benchmarked against HLB: it has the simplest possible binding operations (elementwise multiplication and division), is straightforward to optimize, and achieved the best accuracy in Figure 3. We provide the timed functions at the end of this response.
> > >
> > > OSC's retrieval consists of three matrix multiplications (plus elementwise normalization):
> > >
> > > (1) $(d,k)\times(k,d)\rightarrow(d,d)$
> > >
> > > (2) $(N\cdot p,d)\times(d,d)\rightarrow (N\cdot p,d)$
> > >
> > > (3) $(N,d)\times(d,d)\rightarrow(N,d)$
> > >
> > > GPUs are bottlenecked either by arithmetic throughput (compute-bound) or data movement (memory-bound). Because VSAs require much larger vectors to reach comparable accuracy, their operations become memory-bound: an issue as the GPU spends more time moving data than computing. OSC has a higher FLOP count, but its smaller footprint keeps operations compute-bound, making it faster at scale despite more FLOPs. However, we recognize FLOPs are a useful metric. The FLOPs follow directly from the matrix dimensions, which we will include in the draft alongside the wall-clock timings.
> > >
> > > These larger regimes are quite relevant in practice. As an example, consider the three largest datasets in our XML experiment with vocabularies of 13K, 31K, and 205K (see our response to Reviewer PRJo). There are many datasets with considerably more labels, a few being WikiTitles-500K, ORCAS-800K, and AmazonTitles-3M.
> > >
> > > ```
> > > def time_HLB():
> > >     retrieved   = memory / (role + 1e-9)
> > >     retrieved_n = F.normalize(retrieved, dim=0)
> > >     return cand_normed @ retrieved_n
> > >
> > > def time_OSC():
> > >     R       = torch.eye(d, device="cuda") - role.T @ role        #(1)
> > >     residue = torch.einsum('bpd,dm->bpm', candidates, R)         #(2)
> > >     norms   = residue.norm(dim=-1, keepdim=True).clamp(min=1e-9)
> > >     clean   = residue / norms
> > >     Mv      = clean[:, 1, :] @ M.T                               #(3)
> > >     return (clean[:, 0, :] * Mv).sum(dim=-1)
> > > ```
> > >
> > > **Bullet 4**
> > >
> > > Thank you. Your suggestion is that the paper should make a case for why studying VSA-style superposition memory is justified given attention's empirical success. We will directly address this question below and in the revised draft.
> > >
> > > The main motivation for VSA-style binding is compositional structure. Existing results \[1\] report transformers can achieve high accuracy on compositional tasks seen during training, but fail when tested on novel compositions of the same components. For example, a model can learn "A composed with B" and "C composed with D" but cannot handle "A composed with D" at test time. Interestingly, \[2\] showed that equipping models with explicit TPR binding recovers generalization. A central motivation of the TPR/VSA research program is to address this gap. Binding encodes compositional structure into the representation, capturing nested role-filler relationships in a single vector instead of being spread across an unstructured key-value table. Neural key-value memories have no native mechanism for this recursive nesting. The major practical bottleneck is that while TPRs provide exact binding, they scale exponentially with depth. VSAs compress to fixed size but degrade under superposition. OSC addresses this issue by decoupling structural depth from memory growth. This is a necessary step towards making TPR-style compositional binding usable at scale.
> > >
> > > \[1\] Faith and Fate: Limits of Transformers on Compositionality, NeurIPS 2023
> > >
> > > \[2\] Compositional Generalization Across Distributional Shifts with Sparse Tree Operations, NeurIPS 2024

---

### Official Review · Reviewer_Vozi · 2026-03-13

**Soundness:** 3
**Presentation:** 3
**Significance:** 2
**Originality:** 3
**Overall Recommendation:** 4
**Confidence:** 2

**Summary:**

In this paper, the authors propose Orthogonal Subspace Carving (OSC), a memory architecture for symbolic binding in vector spaces. Instead of expanding the tensor order with structural depth (as typically done in TPRs) or compressing everything into a fixed vector (as done in VSAs), OSC projects filler components onto the null space of a context-defined subspace before combining them via an order-$p$ tensor product. The key claim the authors make is that this decouples structural depth from memory size. Contexts of arbitrary complexity can be represented without increasing the tensor order, and the sub-linear scaling of filler dimension relative to memory capacity yields large storage savings over VSAs in high-superposition regimes. The experimental results compare OSC against VSA baselines on synthetic retrieval tasks and extreme multi-label classification (XML).

**Compliance With Llm Reviewing Policy:**

Affirmed.

**Key Questions For Authors:**

1. Does the total-memory comparison favor OSC because of codebook compression? What happens if you fix the superposition memory size (tensor parameter count) instead? Does OSC still win, or is the advantage purely from the codebook side?

2. TPRs are excluded entirely, but at shallow depth (like depth=1 considered in the synthetic experiments) aren't they feasible and exact? Even a small comparison positioning OSC between TPR and VSA would help, where does the transition happen?

3. How does the retrieval cost of OSC scale with vocabulary size? Will this be a concern for large vocabularies?

4. Have you tried OSC on actual recursive structures (parse trees, programs) at varying depths and compared against sparse TPR approximations?

5. Any intuition for why $k \approx O(\sqrt{d})$ works best for the carving dimension? I feel this seems like an important knob, as setting it too large could make fillers lose distinctiveness, and setting it too small can cause contexts to not separate well. A discussion on this in the main paper would be helpful.

6. For the XML results, do you think fixing a parameter budget and comparing the accuracies show a clearer picture of when OSC's storage advantage helps downstream?


See weaknesses section for more detailed questions and concerns.

**Limitations:**

The paper discusses most of its limitations, but a few more can be added, as detailed in the weaknesses section.

**Strengths And Weaknesses:**

## Strengths

1. The core idea of the paper seems pretty clean and well-motivated to me. The memory triangle tradeoff (fidelity vs. memory footprint vs. superposition) is a useful framing, and the insight that recognition (codebook search) rather than recall (exact unbinding) is what really matters in practice is well-argued.

2. The empirical results on synthetic benchmarks are quite comprehensive, with 14 VSA baselines, multiple dimensions, role depths 1 and 48, and also codebook scaling analysis. The storage efficiency numbers reported are quite impressive (though I have some questions regarding the fairness of the comparisons, in the weaknesses section).

3. The procedural context generation part seems like a nice practical design choice that eliminates the need to explicitly store contexts.

4. The Clifford algebra connection is interesting and provides a geometric interpretation.

## Weaknesses

**1. I don't think the comparison with VSAs is entirely an apples to apples comparison across all resources:** OSC uses an order-$p$ tensor memory ($d^p$ entries) while VSAs use a $d$-dimensional vector. The paper's main efficiency argument is that OSC's *total* memory (tensor + codebook) is smaller than VSAs' total memory for the same retrieval accuracy. This is true and the numbers are convincing, but I think a few things should be noted:
- OSC's memory tensor itself, if I understand correctly,  is $d^p$ entries. So for $d = 200, p = 2$ that's $40{,}000$ parameters, comparable to a VSA at $d = 40{,}000$. I believe the savings would be coming from the codebook, where each filler costs $p \cdot d = 400$ vs. $40{,}000$ for the VSA? So is the advantage specifically/only in settings with large vocabularies and high superposition?
- For small vocabularies or low superposition (where the codebook is a small fraction of total memory), would the advantage shrink? A discussion on this or experiments showing effect of vocabulary size would be useful.
- The paper also excludes comparisons with TPRs from experiments "for structural reasons" (which I understand refers to exponential memory). But for shallow structures (depth 1-2), TPRs are feasible and would be exact. A comparison at shallow depth showing where OSC sits between TPR and VSA would help give a complete picture.

**2. OSC cannot unbind and retrieval requires iterating through the entire codebook:** This is the key limitation with the approach and the paper frames it as "not a functional disadvantage" since VSAs also use cleanup memories. But isn't there a cost difference? If I understand correctly, VSA unbinding gives a noisy vector that can be compared to the codebook in $O(|V| \cdot d)$ time, while OSC retrieval requires constructing a candidate tensor for each vocabulary entry and computing a Frobenius inner product, which naively could cost $O(|V| \cdot d^p)$ per query. The paper doesn't really discuss this computational cost of retrieval, and for large vocabularies, I feel this could be significant.

**3. The XML experiment (Table 2) shows competitiveness but not clear advantage:** OSC ranks first in 7 out of the 16 metrics while VTB ranks first in 6. These are pretty close, and on some datasets (Wiki10, Delicious-200K) OSC underperforms VTB or HLB. The paper states this "confirms that our binding mechanism preserves learnability", which is fair,  but the XML results still don't demonstrate that OSC's storage advantage translates to better downstream performance.

**4. The paper doesn't really validate the deep recursive binding claim experimentally:** One of the key motivations given in the paper is that TPRs scale exponentially with structural depth while OSC doesn't. But there's no experiment testing actual recursive structures  e.g., encoding and retrieving from parse trees of varying depths, or comparing against TPR approximations (like the sparse TPRs of Soulos et al. 2024 mentioned in the intro). This seems like a natural experiment given the paper's framing and its absence makes the claims weak.

**5. There are some presentation issues that could be addressed to improve the paper:**
- The paper should better explain early on that OSC is a recognition-only architecture. The fact that exact unbinding is impossible is a fundamental limitation that readers should know upfront, not discover in Section 4.6.
- The Clifford algebra material is interesting but seems disconnected from the main text. One or two sentences in the main paper explaining what the Clifford formulation buys would help.
- In Figure 3, the methods are a bit hard to distinguish from the plots because of the color scheme used. for eg. GHRR and OSC200 look very similar in color. A better color scheme and visualization of these plots is needed to make the comparison more stark and understandable.
- (This is on a side note or maybe for future extensions) The memory triangle framing is interesting, but I feel its introduced quite informally. Can you think of a way to make it slightly more precise, maybe stating the tradeoff as an impossibility result or a Pareto frontier? I think this would give it more weight from a theoretical standpoint.

---

> ### Author Rebuttal · Authors · 2026-03-31
>
> We appreciate the reviewer's thoughtful engagement with our work. We respond to each question below.
>
> **- Is the advantage only in settings with large vocabularies and high superposition? For small vocabularies or low superposition, would the advantage shrink?**
>
> Yes, the advantage occurs when the vocabulary is large enough. Superposition level doesn't matter for memory savings as the vocabulary must be stored regardless. For the XML experiment, see our response to Reviewer PRJo where we show that for smaller datasets, the total memory size is comparable as your intuition suggests, whereas for larger datasets, OSC uses substantially less memory.
>
> **- Does the total-memory comparison favor OSC because of codebook compression? What happens if you fix the superposition memory size (tensor parameter count) instead?**
>
> Yes, the total-memory comparison favors OSC because of the codebook compression. We explore when superposition memory size is fixed in Figure 5: OSC lags behind HLB (best-performing VSA) by 1-5%. However, the codebook compression gives OSC such a total memory advantage that a larger superposition memory is well within the parameter budget. In Figure 5, OSC uses 1178x less total memory than HLB for comparable accuracy.
>
> **- TPRs are excluded entirely, but at shallow depth (depth=1) aren't they feasible and exact?**
>
> Thank you for the great suggestion! The table below shows minimum dimensions for TPR and OSC to reach 99% accuracy across varying levels of superposition (N) at a depth of 1. OSC achieves this with fewer parameters after a small amount of superposition, and the compression ratio increasingly favors OSC at scale. At larger role depths, TPR's exponential growth quickly becomes infeasible, motivating VSAs. Note TPR is only exact when the dimension is large enough for orthogonality. We will include this comparison, where TPRs are feasible, in the revised draft.
> ||TPR||OSC(p=3)||Compression Ratio|
> |:-|:-:|:-:|:-:|:-:|:-:|
> |N|d|Total Params|d|Total Params|TPR/OSC|
> |50|44|4K|13|4K|1.00x|
> |100|61|10K|16|9K|1.10x|
> |500|137|87K|27|60K|1.45x|
> |1,000|197|236K|34|141K|1.67x|
> |5,000|451|2.5M|59|1.1M|2.25x|
> |10,000|649|6.9M|76|2.7M|2.55x|
>
> **- Will the retrieval cost of OSC be a concern for large vocabularies?**
>
> This is not a concern as OSC's operations are GPU-friendly and its smaller footprint gives a speed advantage over VSA at scale. Even though VSA operations are simpler, VSA computations become memory-bound instead of compute-bound, whereas OSC doesn't suffer from this issue. We provide wall-clock timings and more details in our response to Reviewer ZxPM.
>
> **- Have you tried OSC on recursive structures at varying depths and compared against sparse TPR approximations?**
>
> Our current experiments focus on establishing OSC's capacity and efficiency properties relative to VSAs and validating its learnability in downstream tasks via XML classification. Evaluating OSC on recursive structures and comparing against sparse TPR approximations involves integrating OSC into architectures like the Differentiable Tree Machine, a substantial effort that warrants its own dedicated study, built upon the architecture and codebase this work provides.
>
> **- Any intuition for why $k \approx O(\sqrt{d})$ works best for the carving dimension? Setting it too large could make fillers lose distinctiveness, and setting it too small can cause contexts to not separate well.**
>
> The trade-off involves two competing terms, cross-context interference (favoring large k) and within-context filler discrimination (favoring small k). While our intuition matches yours, pinning down the exact optimum analytically is an open question. However, a core feature of OSC is the ability to increase the tensor rank to keep the filler dimension small, so in practice there is never a large carving dimension to search over. We will add this intuition to the draft, though formally characterizing the optimum remains open.
>
> **- For the XML results, do you think fixing a parameter budget and comparing the accuracies show a clearer picture of when OSC's storage advantage helps downstream?**
>
> Thank you for the question. We would like to clarify the advantage OSC brings to this experiment. OSC achieves competitive accuracy while using substantially fewer parameters, $\sim 5$x less, for the larger vocabularies. We present a table of the parameter counts in our response to Reviewer PRJo.
>
> **- (Side note / future extensions) Can the memory triangle be made more precise as an impossibility result or Pareto frontier?**
>
> We agree that formalizing the memory triangle would strengthen the framing. This is a nontrivial theoretical contribution in its own right, and we believe it deserves a careful treatment beyond what we can provide in this revision.
>
> **- Presentation suggestions**
>
> Thank you for the feedback on the Clifford algebra material, color scheme, and early positioning of OSC as recognition-only. We will update the draft with these helpful suggestions!

---

> > ### Author Rebuttal · Reviewer_Vozi · 2026-04-02
> >
> > Thank you for providing the clarifications and additional results, I believe my concerns have been adequately addressed.

---

### Official Review · Reviewer_PRJo · 2026-03-15

**Soundness:** 3
**Presentation:** 3
**Significance:** 3
**Originality:** 3
**Overall Recommendation:** 4
**Confidence:** 2

**Summary:**

This paper presents a new way in designing a memory architecture called OSC beyond TPR and VSA. Instead of maintaining a memory tensor that grows with context length, or projecting the memory to the original vector space, the paper propose to keep a cartesian product of $p$ vectors and model context binding as projecting to some subspace. The new method supports recognition query but not reproduction. The paper further prove the method is effective through empirical experiments.

**Compliance With Llm Reviewing Policy:**

Affirmed.

**Key Questions For Authors:**

Please refer to the questions in strength and weakness. Some additional question include

1. In the training setup, is the memory size preserved across different methods?

**Limitations:**

yes

**Strengths And Weaknesses:**

**Soundness:** The paper propose a new method with strong theoretical and empirical justification. There is a minor concern on why the paper needs to frame this method as projecting away some of the subspace rather than projecting onto the orthonormal space.

**Presentation:** As someone without a lot of background in this subfield, the reviewer found the authors' introduction on the prior methods and their limitation very clear. There are two small comments

1. On the terminology "SNR" on line 130. This term seems to be "noise to signal ratio" rather than "signal to noise ratio".

2. The authors mentioned in abstract there is a formulation in Clifford algebra, but defer the entire discussion to appendix without any context in the main paper.

**Significance & Originality:** Without too much prior knowledge, the reviewer thinks this is a novel approach that address the drawback of the previous approaches through an algorithmic improvement.

---

> ### Author Rebuttal · Authors · 2026-03-31
>
> We thank the reviewer for taking the time to engage with the work and provide thoughtful feedback. Below we answer your questions.
>
> **- There is a minor concern on why the paper needs to frame this method as projecting away some of the subspace rather than projecting onto the orthonormal space.**
>
> The two framings are indeed interchangeable. We chose the exclusion framing because OSC is defined in terms of the context, and it is more natural to describe the operation as removing the context's contribution than to first characterize the complement and then project onto it.
>
> **- In the training setup, is the memory size preserved across different methods?**
>
> No, we did not constrain total memory to be equal across methods. For smaller datasets, the memory footprints happen to be comparable, but for larger datasets the sub-linear scaling of the filler dimension provides total parameter savings. As shown below, OSC uses substantially less total memory than VSAs on datasets with larger vocabularies.
>
> | | Bibtex | Delicious | Mediamill | Eurlex-4K | Eurlex-4.3K | Wiki10-31K | Amazon-13K | Delicious-200K |
> |:---|---:|---:|---:|---:|---:|---:|---:|---:|
> | **Vocabulary size (# labels)** | 159 | 983 | 101 | 3,993 | 4,108 | 30,938 | 13,330 | 205,443 |
> | **OSC Total memory (# params)** | 55,375 | 261,375 | 40,875 | 1,637,200 | 831,600 | 18,652,800 | 8,088,000 | 123,355,800 |
> | **VSA Total memory (# params)** | 64,000 | 393,600 | 40,800 | 6,390,400 | 1,643,600 | 92,817,000 | 39,993,000 | 616,332,000 |
>
> **- The authors mentioned in abstract there is a formulation in Clifford algebra, but defer the entire discussion to appendix without any context in the main paper.**
>
> We appreciate this feedback! We included the Clifford algebra connection in the abstract to signal its existence to interested readers, but deferred the full treatment to the appendix. Based on past experience, we have found that including Clifford algebra in the main body tends to either alienate readers unfamiliar with the framework or distract from the core contribution. We are happy to add a brief motivating paragraph in the main paper that provides context and points the reader to the appendix for details given the extra page for the final submission.
>
> **- On the terminology "SNR" on line 130. This term seems to be "noise to signal ratio" rather than "signal to noise ratio".**
>
> Thank you for catching this. We have updated the draft.

---

> > ### Author Rebuttal · Reviewer_PRJo · 2026-04-05
> >
> > My concerns are resolved and I will keep my positive rating.

---

### Decision · Program_Chairs · 2026-04-30

**Decision:**

Accept (regular)

**Comment:**

All reviewers agree this work is clearly presented and the proposed method offers a useful trade-off between footprint and fidelity, enabling memory compression at scale with many fewer parameters. Authors have incorporated reviewer feedback on motivation and provided more detailed comparisons to fully address most remaining concerns.